# Clonal chromosomal mosaicism and loss of chromosome Y in elderly men increase vulnerability for SARS-CoV-2
Luis A. Pérez-Jurado [1,2,3,279] ✉, Alejandro Cáceres [4,5,279], Laura Balagué-Dobón [4,5], Tonu Esko [6,7], Miguel López de Heredia [3], Inés Quintela [3,8], Raquel Cruz [3,8,9,10], Pablo Lapunzina [3,11,12], Ángel Carracedo [3,8,9,10,13], SCOURGE Cohort Group* & Juan R. González [4,5,14] ✉

The pandemic caused by severe acute respiratory syndrome coronavirus 2 (SARS-CoV-2, COVID-19) had an estimated overall case fatality ratio of 1.38% (pre-vaccination), being 53% higher in males and increasing exponentially with age. Among 9578 individuals diagnosed with COVID-19 in the SCOURGE study, we found 133 cases (1.42%) with detectable clonal mosaicism for chromosome alterations (mCA) and 226 males (5.08%) with acquired loss of chromosome Y (LOY). Individuals with clonal mosaic events (mCA and/or LOY) showed a 54% increase in the risk of COVID-19 lethality. LOY is associated with transcriptomic biomarkers of immune dysfunction, pro-coagulation activity and cardiovascular risk. Interferon-induced genes involved in the initial immune response to SARS-CoV-2 are also down-regulated in LOY. Thus, mCA and LOY underlie at least part of the sex-biased severity and mortality of COVID-19 in aging patients. Given its potential therapeutic and prognostic relevance, evaluation of clonal mosaicism should be implemented as biomarker of COVID-19 severity in elderly people.

The pandemic of coronavirus disease 2019 (COVID-19) caused by severe acute respiratory syndrome coronavirus 2 (SARS-CoV-2) represented a major health threat to the entire world[1]. During three years, there have been almost 700 million confirmed cases of COVID-19 worldwide, with more than 7 million deaths reported. A best estimate of the overall case fatality ratio after adjusting for demography and under-ascertainment in the initial outbreak in China was 1.38% (95% confidence interval 1.23–1.53), being significantly higher in aging people (6.4% in ≥60 and 13.4% in ≥80 age groups) and in males[2]. World-wide data of the age-stratified case fatality ratio and infection fatality ratio show a similar pattern with a remarkable sex-bias increasing with advanced age, with 60% overall deaths reported in men (estimated hazard ratio of 1.59, 95% confidence interval 1.53–1.65)[3]. Interestingly, sex-dependent differences in disease outcomes were also found during the past SARS-CoV and MERS-CoV epidemics[4,5] and also in mice infected with the virus[6].

Understanding the underlying basis of this different sex and age vulnerability is crucial because aging men and women are likely to have fundamentally different reactions to the SARS-CoV-2 virus infection, treatments, and vaccines. Male patients with COVID-19 have higher plasma levels of innate immune cytokines (IL-8 and IL-18) and stronger induction of non-classical monocytes, while females had more robust T cell activation

during infection. Proposed causes include different case definition of disease, different environmental and social factors (such as lifestyle, smoking history or work-environment) and sex-specific immune-defense factors. The X chromosome harbors multiple genes important for immunity and there are many X-linked immunodeficiencies, so males have greater susceptibility to infections starting at birth[6]. More specifically, SARS viruses use the angiotensin converting enzyme (ACE2), encoded by an X-linked gene, as a receptor to enter and infect ACE-2 expressing cells[1]. Sex variation in the expression of this gene with paradoxically higher expression and higher circulating levels in men than in women has also been proposed as a candidate mechanism[7]. However, ascertainment bias and environmental factors are unlikely to prevail in different populations while the gender-specific immune factors or ACE2 variation would not fully explain the increased risk and sex-divergence with aging. The analysis of previously untreated patients with moderate COVID-19 disease revealed that male patients have higher levels of innate immune cytokines and more robust induction of non-classical monocytes, while female patients have more robust T-cell activation, which is sustained in old age[8]. A B-cell autoimmune disorder present in about 10% of individuals with life-threatening COVID-19 pneumonia has been reported, 5 times more common in males than females, characterized by detection of neutralizing immunoglobulin G autoantibodies against

---

A full list of affiliations appears at the end of the paper. *A list of authors and their affiliations appears at the end of the paper. ✉e-mail: luis.perez@upf.edu; juanr.gonzalez@isglobal.org

interferon type 1[9]. Finally, a meta-analysis of genome-wide association studies searching for host-specific genetic factors has revealed 13 loci significantly associated with SARS-Cov2 infection or severe manifestations of COVID-19, but do not fully explain the gender differences[10].

Mosaic chromosomal alterations (mCA) detectable in blood, including deletions, gains or copy neutral changes, are age-related somatic alterations that indicate clonal hematopoiesis when detectable and have been associated with increased risk for cancer, cardiovascular disease and overall mortality[11–15]. Expanded mCAs have also been recently associated with increased risk for incident infections, including COVID-19 hospitalization[16]. Multiple germline genetic alleles involved in susceptibility to clonally expanded mCA have been identified, with enrichment at regulatory sites for the immune system[16]. In men, mosaic X chromosome monosomy (XCM), acquired by somatic loss of the Y chromosome (LOY), is the most common copy number alteration in male leukocytes, estimated to occur in <2% men under 60 years of age, but exponentially increasing with aging to 15–40% in 70–85 year-old males and >50% at 93 years of age[17]. LOY has also been associated with a wide spectrum of human diseases including cancer, Alzheimer's disease, cardiovascular disease, and reduced overall life expectancy in men[18–21]. Genetic variation in multiple loci is involved in the inherited susceptibility to LOY, which can also be driven by smoking and other environmental exposures[17]. Extreme down-regulation of chromosome Y gene expression mainly driven by genes with X-chromosome homologs that escape X-inactivation seems to be the functional mediator of the reported association between LOY and disease[22,23].

In women, developmental (causing Turner syndrome) or late onset XCM detectable in leukocytes, usually with loss of the inactive X-chromosome, is found with lower frequency than in men but also increasing with age (0.05% in 50-year old; 0.25% in 75-year old)[24]. Females with XCM have an increased risk for autoimmune disease, recurrent viral infections and earlier cardiovascular mortality[25], which is associated with excessive production of pro-inflammatory cytokines (IL-6), decrease in anti-inflammatory cytokines (IL-10, TGF-β) and a lower CD4:CD8 ratio[26].

We have tested here the hypothesis that mCA and XCM/LOY could be underlying factors for the increased severity and mortality of COVID-19 in the elderly and mainly in men. Overall, we have associated clonal mosaicism with a 50% increase in the risk of COVID-19 lethality. We have also correlated LOY in aging males with multiple parameters of cardiovascular dysfunction, and defined the transcriptomic deregulation that underlies disease risks, including signatures of immune system dysfunction and increased coagulation activity. We have finally studied how some of the genes deregulated by LOY are involved in the response to SARS-CoV-2 infection.

## Results

### Higher Covid-19 severity and mortality in males, a sex-bias that increases with aging

Accumulated data on the age-stratified case fatality ratio and infection fatality ratio in a large sample from Spain, show a pattern with a remarkable sex-bias increase with advancing age (Fig. 1). Available reports, mostly based on hospital records, show the same tendency in other countries. COVID-19 lethality, mCA prevalence and LOY prevalence in men, as previously

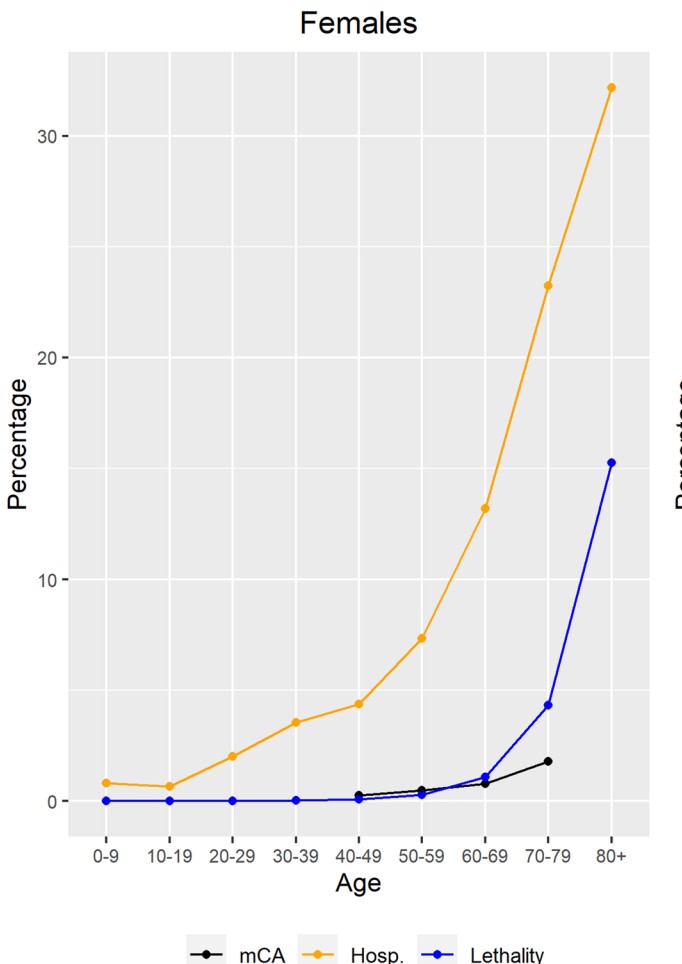

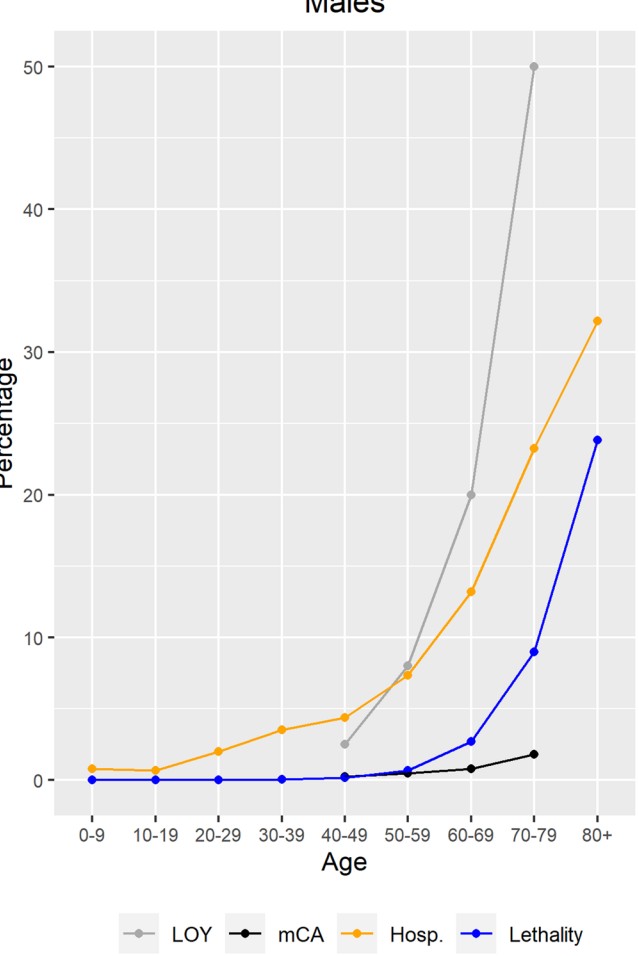

**Fig. 1 | COVID-19 lethality and mCA and LOY prevalence as functions of age.** Increasing sex-specific hospitalization (orange) and mortality (blue) rates for COVID-19 in Spain in the different age intervals (updated July 2022). Estimated prevalence by age in the general population of detectable mCA (black) or LOY in men (grey) in blood is also shown[13,14].

**Table 1 | Number, proportion and mean age of patients in the different clinical categories of COVID-19 severity in the SCOURGE study, with and without detectable mCAs or LOY (males)**

| | | Asymphthomatic (0) | Mild (1) | Moderate (2) | Severe (3) | Critical (4) | Subtotal | No data | Dead[a] | Alive[a] | Subtotal | Total |
|---|---|---|---|---|---|---|---|---|---|---|---|---|
| Cohort (no mCA) | cases | 607 | 2727 | 2141 | 2449 | 1157 | 9081 | 494 | 810 | 8054 | 8864 | 9445 | 9578 |
| | (%) | (6.7) | (30.0) | (23.6) | (27.0) | (12.7) | | | (9.1) | (90.9) | | |
| | age | 53.3 | 51.3 | 66.9 | 70.6 | 65.4 | | | 79.9 | 60.5 | | |
| mCAs | cases | 4 | 18 | 34 | 46 | 17 | 119 | 14 | 29 | 90 | 119 | 133 |
| | (%) | (3.4) | (15.1) | (28.6) | (38.7) | (14.3) | | | (24.4) | (75.6) | | |
| | age | 80.9 | 65.6 | 78.2 | 79.5 | 71.4 | | | 81.1 | 74.1 | | |
| | males | 2 | 7 | 18 | 18 | 15 | 61 | 8 | 18 | 42 | 60 | 69 |
| | (%) | (3.3) | (11.5) | (29.5) | (29.5) | (24.6) | | | (30.0) | (70.0) | | |
| | age | 80.5 | 73.9 | 76.7 | 78.1 | 70.5 | | | 80.7 | 72.1 | | 75.4 |
| | females | 2 | 10 | 16 | 27 | 2 | 58 | 6 | 11 | 48 | 59 | 64 |
| | (%) | (3.4) | (17.2) | (27.6) | (46.6) | (3.4) | | | (18.6) | (81.4) | | |
| | age | 81 | 60.1 | 79.7 | 79.8 | 72.5 | | | 80.6 | 75.4 | | 76.5 |
| Males (no LOY) | cases | 163 | 743 | 1068 | 1241 | 792 | 4007 | 323 | 398 | 3494 | 3892 | 4444 |
| | (%) | (4.1) | (18.5) | (26.7) | (31.0) | (19.8) | | | (10.2) | (89.8) | | |
| | age | 53.2 | 51.9 | 64.8 | 69.1 | 64.0 | | | 77.2 | 61.5 | | |
| Males LOY | cases | 4 | 13 | 47 | 97 | 47 | 208 | 18 | 65 | 143 | 208 | 226 |
| | (%) | (1.9) | (6.3) | (22.6) | (46.6) | (22.6) | | | (31.3) | (68.8) | | |
| | age | 81.2 | 78.5 | 81.0 | 83.5 | 79.3 | | | 84.1 | 81.5 | | |

[a]>90 days after COVID-19.

reported in multiple reports including the UK biobank dataset, appear to increase exponentially with age (Fig. 1)[18–21].

## COVID-19 severity variables and their association with age

We first studied the SCOURGE clinical data. Phenotype data was available from all 9578 individuals (5134 females and 4444 males) diagnosed with COVID-19 and recruited to the SCOURGE study (Table 1). According to disease severity, there were 607 cases asymptomatic (6.8% A), 2727 individuals with mild symptoms (30% L), 2141 patients with moderate disease (23.6% M), 2449 with severe manifestations (27% G), 1157 critical (12.7% C). We visually inspected the contrasts defined together with the level of severity and the age of the patients. Mean age was 62.58 years, 64.34 for males and 61.06 for females, with an age difference between sexes that was statistically significant ($P = 4.1 \times 10^{-19}$). All clinical categories and variables correlated with age except for "critical" and "history of pulmonary thromboembolism".

## Association between mCA and COVID-19 severity

The algorithm followed by manual curation finally detected 133 individuals (1.42%), 61 males and 72 females, carrying mCAs in blood affecting the autosomes and/or the X chromosome (Table 1, Supplementary Data 1, Fig. 2a & 2b and Figure S1). Globally, 95 individuals had a single mCA while 38 of them had more than one event, for a total of 213 mCAs. There were 88 deletions, 5 whole chromosome monosomies, 20 segmental gains and 21 whole chromosome trisomies, along with 78 copy-neutral changes (somatic segmental uniparental disomies), and a few complex rearrangements. Mean age for individuals with mCAs was 75.04 ± 12.7. We then performed association analyses across the different outcome variables related to COVID-19 severity and the presence of mosaicism. We first confirmed the strong association between mosaicism and age (year) ($OR = 1.051$, $P = 1.05 \times 10^{-16}$), as previously reported. We then observed a significant association between the presence of mCA and COVID-19 lethality (1-survival, $OR = 1.75$, $P = 0.015$), after adjusting for sex and age (Fig. 3). The contribution of mCAs to COVID 19 lethality was stronger and more significant in males only (OR 2.16; 95%CI: 1.19-3.93). Although in the same direction, the split sample size was not enough to achieve statistical significance for an association in females (OR 1.32; 95%CI: 0.66-2.67).

## Association between LOY and COVID-19 severity

Among all male cases, we detected 226 individuals with LOY (mean age 82.0 ± 7.9), a 5,08% prevalence of LOY in this cohort (Table 1, Supplementary Data 2), which is within the reported range in previous publications[17–20], albeit significantly below the prevalence reported by others in the UK biobank using the same threshold[21]. The reasons for this discrepancy are unclear, but our calling method of LOY has demonstrated robustness in comparison with others, using both simulations and real data[27]. According to the estimated proportion of cells with XCM or LOY, 162 individuals had mild LOY (<25% cells with LOY), 43 moderate LOY (25-65% cells with LOY) and 21 had extreme LOY (>65% cells with LOY) (Fig. 2c). We also identified three women with detectable chromosome Y in a proportion of cells, then likely corresponding to X0/XY mosaicism and a possible diagnosis of Turner syndrome, as well as three individuals with non-mosaic XYY (Fig. 2c). We observed 6 men with both LOY and mCA, 220 with LOY and no mCA, and 55 with mCA and no LOY, which resulted in no significant correlation between the presence of LOY and mCA.

We first confirmed a strong association between LOY in males and age ($OR = 1.11$, $P = 5.65 \times 10^{-51}$). We then fitted a series of models between LOY and the contrast $C \text{ or } G > M \text{ or } L \text{ or } A$, for which we had observed a strong association with age. We first observed a significant association between the contrast and LOY, primarily due to its association with age (not significant after adjusting by age, $OR = 1.25$, $P = 0.15$). We also performed association tests for all the contrasts and clinical variables adjusting only by age and we observed some significant associations. LOY was associated with reduction in survival ($OR = 0.713$, $P = 0.045$) and with clinical history of vascular disease ($OR = 0.627$, $P = 0.001$) and lung thromboembolism ($OR = 0.271$, $P = 0.042$). While associations with severity were not significant, we observed a consistent estimate of their risk given by LOY.

We then tested the association with the continuous value for mLRRY across all severity contrast and clinical variables. We found a significant association with survival for higher relative levels of chromosome Y content ($\beta = 0.86$, $P = 0.0054$).

We then performed a joint analysis for all mosaicisms, mCAs and LOY, confirming their strong association with age ($OR = 1.08$ $P = 1.95 \times 10^{-62}$) and with COVID-19 lethality ($OR = 1.53$, $P = 0.004$) after corrections, including adjustment for other clinical variables (Fig. 3). The associations of

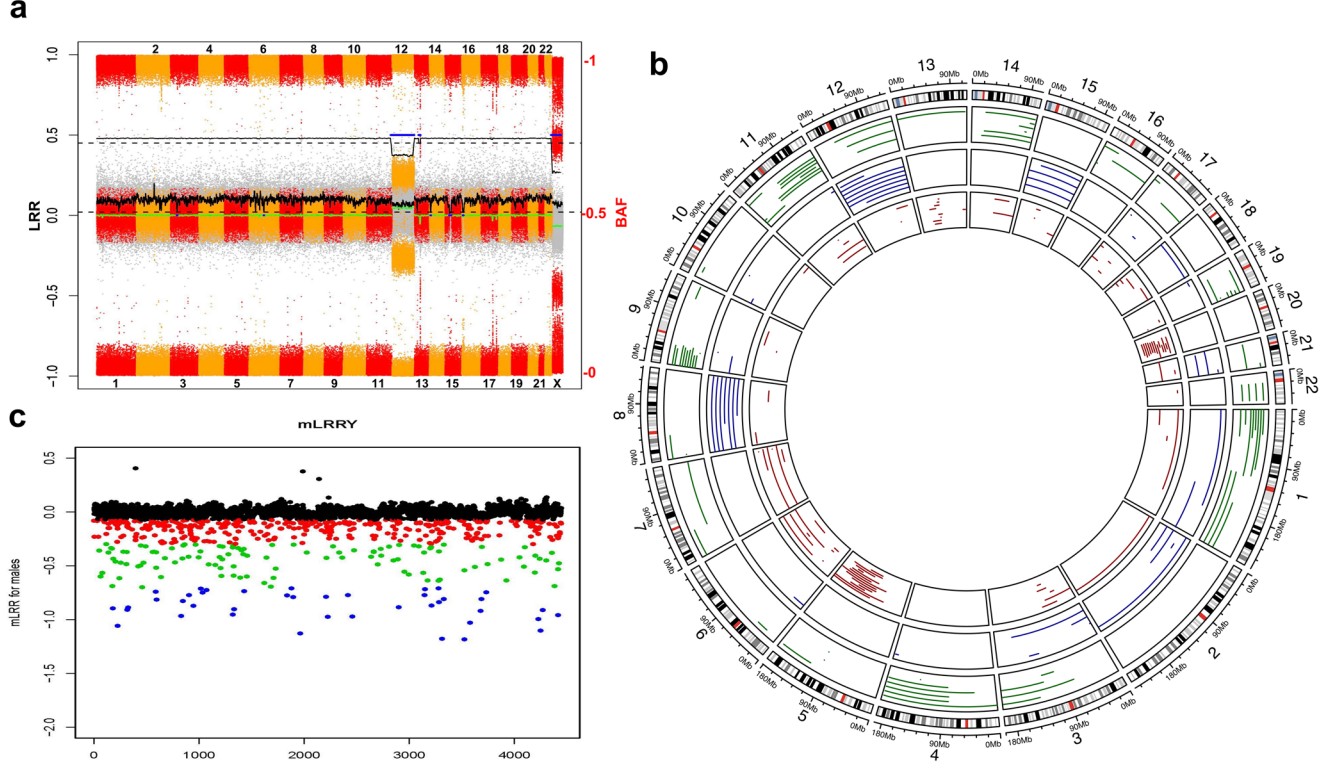

**Fig. 2 | mCA and LOY detection if the SCOURGE study. a** Plot representing the whole-genome molecular karyotype obtained by SNParray of blood DNA from an individual with several mCAs. Dots in grey are LRR values (average per widow shown by a green lane), while colored dots are BAF values of homozygous and heterozygous SNPs from odd (red) and even number (orange) chromosomes, respectively. Abnormal BAF and average LRR values in three regions (blue lanes interrupting the black lane in the upper part) correspond to mosaicism for trisomy 12, a small interstitial deletion in 13q and X-chromosome monosomy. The blue lanes interrupting the green lane at LRR = 0 correspond to small regions of homozygosity.

**b** Circus plots showing all detected mCAs in the SCOURGE dataset. In red deletions, in blue gains and in green copy neutral events. **c** Analysis of LOY in male individuals in the SCOURGE study based on mean LRR from chromosome Y (mLRRY: relative amount of DNA from the Y chromosome with respect to autosomes). Blue dots correspond to males with mosaic LOY in more than 65% of cells (XCM > 65%), green dots to males with LOY/XCM between 25%–65%, and red dots to males with LOY/XCM in less than 25% of cells. The three individuals with top mLRRY values have apparently non-mosaic gains of chromosome Y (47,XYY).

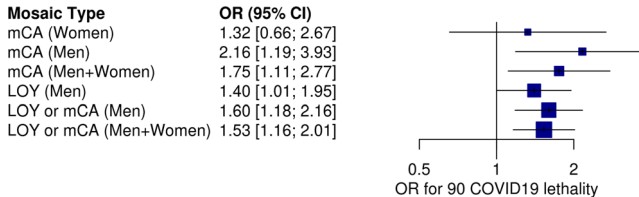

| Mosaic Type | OR (95% CI) |
|---|---|
| mCA (Women) | 1.32 [0.66; 2.67] |
| mCA (Men) | 2.16 [1.19; 3.93] |
| mCA (Men+Women) | 1.75 [1.11; 2.77] |
| LOY (Men) | 1.40 [1.01; 1.95] |
| LOY or mCA (Men) | 1.60 [1.18; 2.16] |
| LOY or mCA (Men+Women) | 1.53 [1.16; 2.01] |

**Fig. 3 | Associations of detectable mCAs and LOY with COVID-19-related mortality.** Mortality was reported at most 90 days after infection. The analyses are stratified or adjusted by sex, as indicated. All analyses are adjusted for age and 10 principal components of ancestry. Individuals with prevalent hematologic cancer were excluded from the analysis. Error bars correspond to the 95% CI for the OR estimates.

all type mosaicism with severity contrasts were not significant but consistent across all contrasts.

## Germline aneuploidies and COVID-19
In addition to 6 individuals with XCM and likely Turner syndrome, 3 cases with 45,X0/46,XY mosaicism mentioned above, 2 more cases with 45,X0/46,XX mosaicism and one with likely 45,X0/46,XY/46,XX mosaicism, the algorithm also detected a total of 25 individuals with germline (non-mosaic) aneuploidies. We detected 7 cases with Down syndrome (trisomy 21) and 18 with gonosomal aneuploidies, including 9 with Klinefelter syndrome (47,XXY), 6 with triple X syndrome (47,XXX) and 3 with XYY syndrome (47,XYY) (Supplementary Data 3). We found an association of aneuploidies with the presence of mCAs (OR = 9.90, P = 0.0047).

We then performed association tests of phenotypic features with all the aneuploidies, removing individuals with mCAs. We did not find any significant association between COVID-19 severity parameters and any type of aneuploidy given this small sample size, although previous history of cardiopathy was significantly associated, as expected (OR = 4.02, P = 0.004).

## Correlation of LOY with cellular and biochemical phenotypes in EGCUT individuals
We analyzed SNP microarray data with MADloy of a selected sample of 530 apparently healthy adult men from the Estonian Genome Center of the University of Tartu cohort (EGCUT) and classified them as having (n = 28) or not having LOY (n = 502). We then correlated genotype classification with several clinical parameters. Individuals with LOY had significantly age-adjusted decrease in red cell counts, decrease in mean corpuscular hemoglobin concentration and higher red cell distribution width, low basophil counts and borderline low lymphocyte proportions. Biochemical parameters revealed low albumin levels, low triglycerides and elevated homocysteine and urea levels (Table S1).

## Blood transcriptome in individuals with LOY reveals immune defects and cardiovascular risk
We also compared blood transcriptome from 11 men with LOY (median age: 69, range: 58-84) and 32 age-paired men without LOY (median age: 68, range: 60-87) as controls. Multiple genes differentially expressed between groups were found, including autosomal and gonosomal genes (Tables S2, S3 and Figures S2, S3), providing insight into the mechanisms of disease

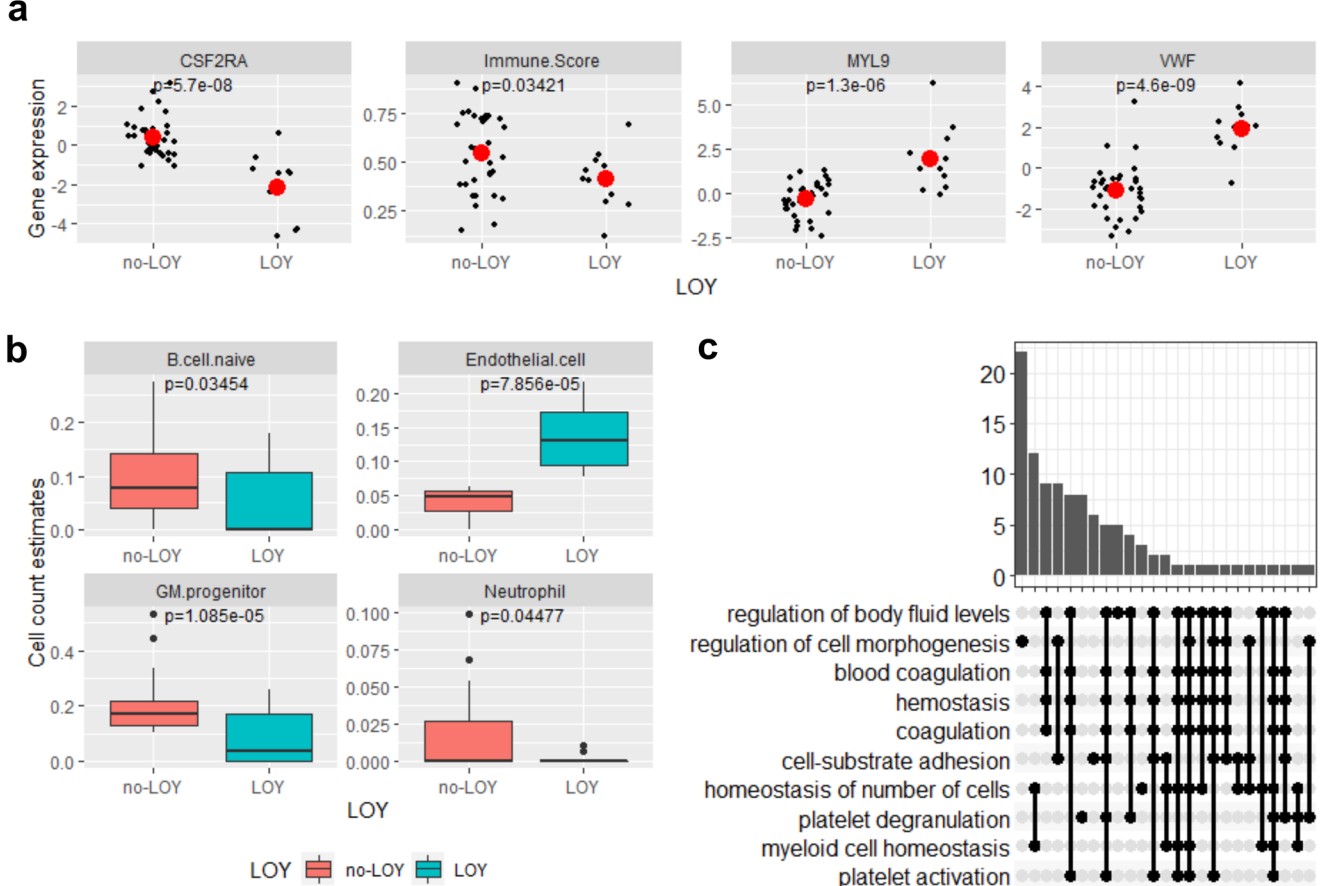

**Fig. 4 | Transcriptomic signatures of LOY. a** Decreased expression of *CSF2RA* mRNA and increased expression of *MYL9* and *VWF* in individuals with LOY compared with controls with no LOY (mean gene expression in red dot). **b** Different predicted cell counts underlying the transcriptomic differences between cases with LOY and control individuals (no-LOY). **c** Gene Ontology (GO) enrichment of top differentially expressed genes. Boxplots show the interquartile range, while error bars represent the spread of data around the median.

susceptibility caused by LOY with implications for COVID-19. *CSF2RA*, located on the X-Y chromosome pseudoautosomal 1 (PAR1) region, is one of the most significantly down-regulated genes in LOY (Fig. 4a), along with other multiple Y chromosome genes with homologs on the X chromosome that escape X inactivation and with known function in immunity (Supplementary Data 4 and 5).

Top autosomal genes overexpressed in LOY, such as *VWF and MYL9* (Fig. 4a), are associated with cardiovascular risk. *VWF* codes for the von Willebrand factor (vWF), a pro-coagulant protein that promotes platelet adhesion and smooth muscle cell proliferation, while *MYL9* encodes Myosin Light Chain 9, regulatory, important in inflammatory immune responses.

Since changes in gene expression may reflect differences in cell-type composition and functionality, we estimated the average cell-type functional composition of samples from individuals with LOY compared to those without LOY using bulk transcriptome data (Table S4). The results were consistent with LOY individuals having significantly decreased GM-progenitors and B cell naïve cells, along with increased counts of endothelial cells (Fig. 4b). Enrichment gene set analysis using differentially expressed genes revealed a few categories significantly over-enriched, most notably the coagulation and cellular detoxification, the leukocyte migration and neutrophil activation (Fig. 4c, Tables S5 and S6). Overall, gene expression in LOY individuals leads to a down-regulated immune score.

### Blood transcriptome in individuals with mCAs

For transcriptome analysis, we selected individuals only with copy-neutral mCAs to minimize the variability secondary to the individual rearrangements. We then compared blood transcriptome from the 9 individuals with mCA and 90 age-paired individuals without mCA as controls. A total of 83 genes differentially expressed between groups were found, but enrichment analysis did not reveal any significantly deregulated pathway (Supplementary Data 6).

### Down-regulated genes in LOY involved in response to SARS-CoV-2 infection

We tested whether the genes that participate in the primary response to SARS-CoV-2 infection were significantly deregulated in blood cells of individuals with LOY. We obtained 249 deregulated genes with SARS-CoV-2 infection in primary human lung epithelium (NHBE) and 130 for transformed lung alveolar (A549) (339 unique genes for the two cell lines). This gene set is highly over-represented in several pathways including defense response to virus, IL-17, type I interferon and NF-Kappa B signaling (Table S7). From the deregulated genes in cells infected with SARS-CoV-2, 13 were also deregulated in individuals with LOY (Fig. 5a and Table S8) indicating a strong significant over-representation (OR of enrichment = 7.23, $p = 1.5 \times 10^{-7}$). Most of these genes are interferon response genes (*IFIT3, IFI44L, ITFT1, IFI6*), which are down-regulated in individuals with LOY (Fig. 5b–d).

### Discussion

We have shown in the SCOURGE study that clonal detectable mCAs, including XCM, are relatively common in blood of aging individuals, as previously reported[14], with much higher frequency in males due to somatic LOY[18]. In addition to a risk factor for cancer, cardiovascular complications,

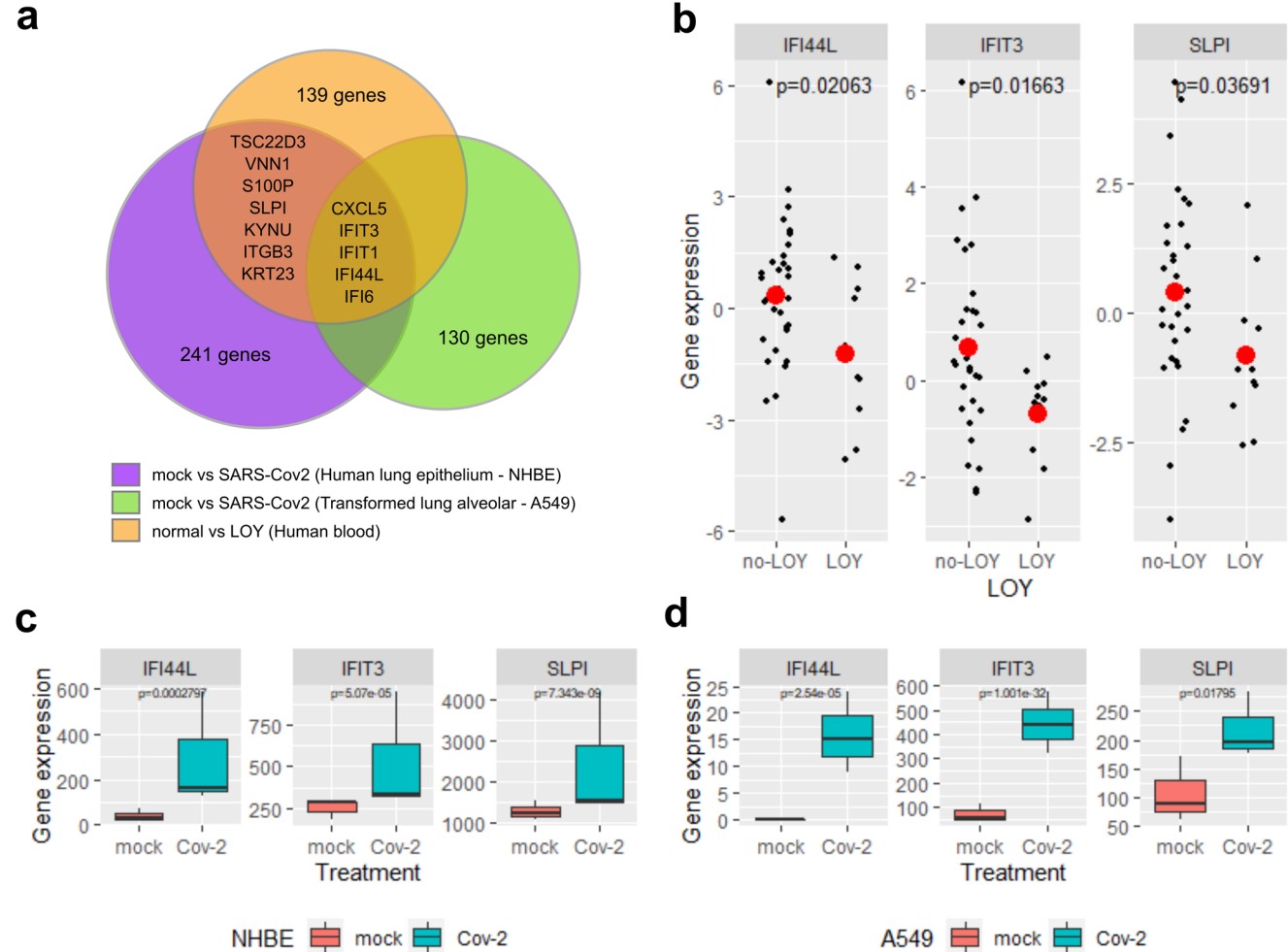

**Fig. 5 | Transcriptomic overlap between LOY and SARS-CoV-2. a** Overlap between top differentially expressed genes in individuals with LOY and deregulated genes in SARS-CoV-2 infected cells. Panels **b**, **c** and **d** show detailed gene expression patterns of some of these overlapping genes, including down-regulated in individuals with LOY (**b**), and over-expressed in NHBE (**c**) and A549 (**d**) cell lines infected with SARS-CoV-2. Boxplots show the interquartile range, while error bars represent the spread of data around the median.

incidental infections and all cause early mortality[12,16,18,28], clonal hematopoiesis with mCA and/or XCM due to LOY is a risk factor for COVID-19 lethality with a combined odds ratio of 1.53. Despite some limitations of our study due to a relatively small sample size and the possibility of uncontrolled confounding factors, similar results have been recently reported in the UK biobank revealing increased risk for diverse incident infections and COVID-19 hospitalization in people with clonal hematopoiesis with chromosomal mosaicism[16,28]. Our data indicate that these two types of chromosomal mosaicism underlie at least part of the aging-related and sex-biased severity and mortality of COVID-19. Therefore, identification of mCA and LOY in blood cells is likely to have an immediate clinical relevance in the management of aged patients with COVID-19.

Clonal hematopoiesis of indeterminate potential due to expansion of peripheral blood cells with acquired point mutations in a specific set of genes, is a similar condition that also increases with age and associates increased risk for cancer, cardiovascular disease, and decreased overall survival[29]. A significant overlap after adjusting for age has been proven between detectable mCAs or LOY and mosaic gene mutations, suggesting a possible synergistic relationship between clonal hematopoiesis with gene mutations and acquired chromosomal rearrangements[30]. Interestingly, a relationship between clonal hematopoiesis with gene mutations and risk of severe infections, including severe COVID-19, has also been recently documented[31]. However, although the increased severity and mortality of COVID-19 could be explained in part by the co-existence of clonal

hematopoiesis, the underlying mechanisms in individuals with LOY are likely different from those in individuals with mCAs or mosaic gene mutations.

The predisposing factors to autosomal events and LOY seem to be mostly unrelated, as no significant association has been found between both types of events in our cohort, their blood transcriptomic signatures do not overlap and the germline loci reported to predispose to autosomal mCAs and LOY are different[16,19,21]. While only 10% of autosomal mCAs correspond to whole chromosome aneuploidies (mainly trisomies 8, 12 and 15 and monosomy 7) likely mediated by mitotic non-disjunction, this is the main mechanism for XCM and LOY. Mitotic non-disjunction of sister chromatids of the Y chromosome may be facilitated by the higher rate of cellular turnover of aging men. In mice, while the Y chromosome is stably transmitted during meiotic cell divisions, there is a high frequency of non-disjunction in mitosis, mainly in the earliest cleavage divisions[32].

A possible pathogenetic mechanism that could be common to clonal mCAs and XCM is immunosenescence, which involves modifications of humoral and cellular immunity. One aspect of immunosenescence is a decline in the absolute number of peripheral blood lymphocytes with locus-dependent reduction of HLA class-I cell surface expression, related with increased risk of subsequent mortality[33]. T-lymphocytes also play a central role in the effector and regulatory mechanisms of the adaptive immune response[34].

Many of the biochemical and transcriptomic alterations found in individuals with LOY have been already associated to poor prognosis for SARS-CoV-2 infection[35,36]. Several genes located on the Y chromosome with relevant functions in the immune system have functional homologs on the X chromosome that escape X inactivation in females (Supplementary Data 4). Cells with XCM are likely haploinsufficient for many of those genes, which are downregulated in individuals with mosaic XCM due to LOY. In this regard, we observed low expression of *CSF2RA* in individuals with LOY, who also have low GM progenitors. *CSF2RA* codes for the alpha subunit of the heterodimeric receptor for colony stimulating factor 2, a cytokine that regulates the production, differentiation, and function of granulocytes and macrophages (GM-CSF), key cells for antigen presentation in infections, and is also critical for T cell function. GM-CSF increases IL-2R and IL-2 signaling, which can increase expansion of lymphocytes and IFN-γ production important for anti-viral response. Therefore, GM-CSF leads to enhanced protective responses[37]. Loss or inactivation of both copies of the *CSF2RA* gene is associated with surfactant metabolism dysfunction-4 and pulmonary alveolar proteinosis, a primary immunodeficiency (OMIM 300770)[38]. As Leukine® (sargramostim, rhu-GM-CSF) has being assessed in the SARPAC trial because of its potential positive effect on antiviral immunity and contribution to restore immune homeostasis in the lungs with inconclusive results (https://clinicaltrials.gov/ct2/show/NCT04326920), our data suggest that patients with LOY might be predictive of a poor response due to their low expression of one of the receptor subunits for GM-CSF (*CSF2RA*)[39].

Patients severely affected with COVID-19 have lower lymphocyte counts, especially T cells, higher leukocyte counts and neutrophil-lymphocyte-ratio, lower percentages of monocytes, eosinophils, and basophils, along with generally elevated levels of infection-related biomarkers and inflammatory cytokines, including IL-6. Helper, suppressor and regulatory T cells were all below normal levels in the severe group, with increased naïve helper T cells and decreased memory helper T cells[1,40]. We observed a significant overlap of deregulated genes in LOY individuals that participate in the immediate immune response elicited by SARS-CoV-2 virus infection. Some of these genes clearly activated in both studied cell types infected by SARS-CoV-2 are markedly under expressed in individuals with LOY (*SLPI, IFI6, IFIT1, IFIT3,* and *IFI44L*) (Fig. 5b–d). Secretory leukocyte protease inhibitor (SLPI) is a regulator of innate and adaptive immunity that protects the host from excessive inflammation in infectious disease, while the other four genes encode interferon induced proteins of the innate immune system that participate in the immediate host response to viral infections[41]. Dysfunctions of the adaptive immunity and interferon-mediated immediate host response in individuals with LOY are consistent with the observed sexual dimorphism in human immune system aging, and might underlie a poor immune response to SARS-CoV-2 infection[42]. This patterns along with the increased severity in older males, suggests that XCM due to LOY may be one underlying factor for susceptibility to COVID-19 in a proportion of patients.

In addition to depleted hematopoietic progenitor cells and possible immunodeficiency, individuals with LOY may have increased levels of circulating endothelial cells, which are known biomarkers for endothelial dysfunction and cardiovascular disease[43]. In a mouse model with LOY, macrophages recruited to the heart showed aberrant profibrotic differentiation leading to cardiac fibrosis during aging[44]. We observed upregulation of *VWF* and *MYL9* in LOY. Pro-coagulant vWF promotes platelet adhesion and smooth muscle cell proliferation, and elevated levels of vWF have been associated with higher risk for thrombosis and cardiovascular disease[45]. MYL9 is a ligand for CD69 to form a net-like structure inside blood vessels in inflamed lungs and is also a risk factor for cardiovascular disease risk found over-expressed in aged versus young injured arteries[46]. Through these mechanisms, LOY seems to contribute to COVID-19 lethality by its associated cardiovascular risk.

In summary, clonal detectable mCA & XCM are relatively common in aging individuals with much higher frequency in males due to somatic LOY. LOY is associated to decreased progenitors and stem cells, along with immune system dysfunction and increased coagulation and cardiovascular risk, as revealed by biochemical and gene expression data. Our data indicate that this type of chromosomal mosaicism underlies at least part of the sex-biased severity and mortality of COVID-19 in aging patients. Given its potential relevance for modulating prognosis, therapeutic intervention, and immunization responses, we propose that evaluation of mCA/LOY by currently established methods should be implemented in both, retrospective studies and all prospective and currently ongoing clinical trials with different medications and vaccines for COVID-19. Testing for mCA/LOY at large scale in elderly people may also be helpful to evaluate vaccination response and to identify still unexposed people who may be especially vulnerable to severe COVID-19 disease.

## Methods

### Covid-19 infection, mortality data, mCA and LOY prevalence estimates

Accumulated data was obtained from the Spanish National Epidemiological Registry (https://www.isciii.es/QueHacemos/Servicios/VigilanciaSaludPublicaRENAVE/EnfermedadesTransmisibles/Paginas/InformesCOVID-19.aspx). Hospitalization rates, intensive care admission rates, and mortality stratified by age and sex was obtained from this report. Prevalence estimates of mCA and LOY by age were obtained from the general population[14,21].

### EGCUT subjects, phenotype and genotype data

LOY and mCA were assessed in a total of 882 adult individuals belonging to the Estonian Gene Expression Cohort (EGCUT, www.biobank.ee) that comprises a large cohort of 53,000 samples of the Estonian Genome Center Biobank, University of Tartu[47]. Detailed phenotypic information from all the individuals studied, including clinical analysis (blood cell counts and general biochemistry) and follow-up until June 2020, was available in ICD-10 codes. Patients selected in this study were genotyped using OmniX array. All individuals had genotyping success rate above 95%. All studies were performed in accordance with the ethical standards of the responsible committee on human experimentation, and with proper informed consent from all individuals tested.

### SCOURGE subjects, phenotype and genotype data

A total of 9578 (5134 females and 4444 males) patients diagnosed with COVID-19 and recruited to the SCOURGE study were included in this study[48]. Mean age was 62.58 years, 61.06 for females and 64.34 for males. Available phenotype data included age, sex, some clinical variables of past clinical history, several defined measures of COVID-19 severity and vital status (alive or dead) 90 days after diagnosis. The severity variables classified individuals in five levels called Asymptomatic (A), Mild (light: L), Moderate (M), Severe (G), and Critical (C). Additional information about pre-existing conditions as categorical variables was also available for most cases, including history of vascular disorders, cardiac problems, neurologic conditions, gastrointestinal disorders, onco-hematologic conditions, respiratory issues, and pulmonary thrombo-embolism. Blood DNA was genotyped using a customized Affymetrix SNP microarray[48]. Genotype data passed quality controls for GWAS analysis. The whole SCOURGE project was approved by the Galician Ethical Committee Ref 2020/197, along with the Ethics and Scientific Committees of all participating centers.

### Detection of mCA and LOY

The genotype CEL files from everyone were used to extract the log-R ratio (LRR) and B-allele (BAF) frequency from SNP probes. We used the *apt* software for quality control (QC) and the extraction of the array intensity signals. Following the QC pipeline with filters *axiom-dishqc-DQC > 0.82* and *call-rate > 0.97*, we observed that all individuals could be included. The signals were obtained from CNV calling pipeline with default parameters *mapd-max = 0.35* and *waviness-sd-max = 0.1*. We also called mosaicisms in autosomes and chromosome X with the MAD algorithm[49]. The method uses the fixed deviation from the expected BAF value of 0.5 for heterozygous SNPs (Bdev) to call allelic imbalances by using a segmentation procedure.

The segmentation was performed using the three different parameters of MAD: $T > 8$, aAlpha = 0.8, minSegLength >100. Some false positive alterations were detected in bad quality arrays. Therefore, curation via visual inspection, considering variability of LRR and BAF mean values in the segment, was performed by two independent investigators. Each mosaic alteration was classified as copy-loss, copy-gain or copy-neutral. The estimated percentage of abnormal cells was computed based on the B-deviation as previously reported[10].

Mosaic LOY detection and quantification was performed using the *MADloy* tool which implements LOY calling using the mean LRR (mLRRY) and B-deviation derived-measures from chromosome Y across subjects[50]. For each sample, *MADloy* first estimates the normalized mLRRY given by its ratio with the trimmed-mean of mLRRY values in the autosomes to discard regions with copy number alterations. B-deviation is calculated for the pseudoautosomal regions 1 and 2 (PAR1, 0–2.5 Mb on both Xp and Yp; PAR2, 300 kb on distal Xq and Yq, Mb 155 and 59, respectively), and the XY transposed region (88–92 Mb on X, 2.5–6.5 Mb on Y). The method is calibrated to detect mosaicism when the proportion of affected cells is above 10%. We then plotted the values of the mLRRY signals for males and females. A signal from chromosome Y in females is observed due to the background noise of the array and some cross-hybridization. While we observed variability of the mLRRY signal, numerous males were identified with extreme low values of mLLRY, suggesting loss of chromosome Y. We categorized the level of LOY status into three groups according to the magnitude of the decrease in mLRRY, believed to be a function of XCM/LOY cellularity.

### Bulk transcriptome data
Gene expression was obtained with Illumina whole-genome expression BeadChips (HT12v3) from peripheral blood RNA in the EGCUT cohort. Low quality samples were excluded. All probes with primer polymorphisms were left out, leaving 34,282 probes. The expression dataset is publicly available at GEO (Gene Expression Omnibus) under the accession number GSE48348[45]. In this dataset, a total of 11 individuals with LOY and 9 individuals with copy-neutral mCAs were identified. In order to consider the effect of aging on LOY/mCA detection and to have the maximum power, 32 age and gender-paired normal samples without LOY or mCA (3 controls per case) were selected for the transcriptomic analyses.

The effect of SARS-CoV-2 infection on gene expression was assessed in independent biological triplicates of two different cell lines that were mock treated or infected with SARS-CoV-2 (USA-WA1/2020). One corresponds to primary human lung epithelium (NHBE) and the other to transformed lung alveolar cells (A549). These data are available at GEO under the accession number GSE147507.

### Statistical data analyses
Gene expression data was quantile-normalized to the median. We analyzed linear regression residuals of gene expression data on forty multi-dimensional scaling components, to correct for possible unwanted variability. Array quality was assessed using *arrayQualityMetrics* Bioconductor package. *genefilter* Bioconductor package was used to filter for features without annotation and/or exhibiting little variation and low signal across samples, leaving a total of 15,592 probes from 34,282. Differential expression (DE) between individuals with and without LOY was then performed using *limma* Bioconductor package. Significant DE genes were considered at false discovery rate (FDR) < 0.05. Significant DE genes at $p < 0.001$ level was selected for Gene Ontology (GO) and KEGG (Kyoto Encyclopedia of Genes and Genomes) enrichment analysis with *clusterProfiler* Bioconductor package. Over-representation of DE genes in the gene set obtained from the analysis of SARS-CoV-2 infected cell lines ($p < 0.001$ and log-foldchange > 0.5) was performed using exact Fisher test. Cell-type composition of the 43 individuals with bulk transcriptomic data (11 LOY, 32 normal) was estimated using the 'xcell' method implemented in the *immunodeconv* R package[51].

Association analysis between mCA or LOY status and clinical data, including blood cell counts and biochemical parameters, was assessed using linear models adjusted by age. All statistical analyses were performed using the statistical software R version 3.6.3 (http://www.r-project.org).

### Reporting summary
Further information on research design is available in the Nature Portfolio Reporting Summary linked to this article.

## Data availability
Transcriptomic data from EGCUT individuals are available at GEO repository under the accession number GSE48348. LOY and mCA status obtained from genomic data are available upon request.

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

## Acknowledgements

We thank Gemma Moncunill, Lluís Armengol and Jozef Gecz for their critical reading of initial versions of this manuscript. The authors acknowledge support from the Catalan Department of Economy and Knowledge (SGR2017/1974, SGR2017/801), the Spanish Ministry of Science "Programa de Excelencia María de Maeztu" (MDM-2014-0370) and "Centro de Excelencia Severo Ochoa" (CEX2018-000806-S), the Fondo Europeo de Desarrollo Regional, UE (RTI2018-100789-B-I00) and the Estonian Research Council (PUT1660). The SCOURGE study has been funded by Instituto de Salud Carlos III (COV20_00622) and ofounded by European Union (ERDF) "A way of making Europe"; additional funding was received from Amancio Ortega Foundation and Banco de Santander. Authors also receive support from the Generalitat de Catalunya through the CERCA Program.

## Author contributions

Conceptualization, L.A.P.J. and J.R.G.; Funding acquisition, L.A.P.J., TE (EGCUT), P.L. and Acar (SCOURGE) and J.RG; Data provider, T.E., M.L.d.H., I.Q., R.C., P.L., Acar, and SCOURGE Cohort Group; Data analysis: J.R.G., Acac, L.B.D. and L.A.P.J.; Methodology, Acac and JRG; Project administration, L.A.P.J., P.L. and J.R.G.; Software, L.B.D. and J.R.G.; Validation, L.A.P.J. and Acac; Supervision and visualization, L.A.P.J. and J.R.G.; Writing original draft, L.A.P.J.; Writing review & editing, L.A.P.J., Acac and J.R.G.. L.A.P.J., Acac and J.R.G. contributed equally to this work. All authors have read and agreed to the published version of the manuscript.

## Competing interests

LAPJ is founding partner and scientific advisor of qGenomics laboratories. The other authors declare no competing interests.

## Ethics approval and consent to participate

All studies were performed in accordance with the ethical standards of the responsible committee on human experimentation, and with proper informed consent from all individuals tested.

## Additional information

[1]Genetics Unit, Department of Medicine and Life Sciences, Universitat Pompeu Fabra, Barcelona, Spain. [2]Genetics Service, Hospital del Mar & Hospital del Mar Research Institute (IMIM), Barcelona, Spain. [3]Centro de Investigación Biomédica en Red de Enfermedades Raras (CIBERER), ISCIII, Barcelona, Spain. [4]Barcelona Institute for Global Health (ISGlobal), Barcelona, Spain. [5]Centro de Investigación Biomédica en Red en Epidemiología y Salud Pública (CIBERESP), Barcelona, Spain. [6]Estonian Genome Science Centre, University of Tartu, Tartu, Estonia. [7]Program in Medical and Population Genetics, Broad Institute, Cambridge, MA, USA. [8]Centro Nacional de Genotipado (CEGEN), Universidade de Santiago de Compostela, Santiago de Compostela, Spain. [9]Instituto de Investigación Sanitaria de Santiago (IDIS), Santiago de Compostela, Spain. [10]Centro Singular de Investigación en Medicina Molecular y Enfermedades Crónicas (CIMUS), Universidade de Santiago de Compostela, Santiago de Compostela, Spain. [11]Instituto de Genética Médica y Molecular (INGEMM), Hospital Universitario La Paz-IDIPAZ, Madrid, Spain. [12]ERN-ITHACA-European Reference Network, Paris, France. [13]Fundación Pública Galega de Medicina Xenómica, Sistema Galego de Saúde (SERGAS), Santiago de Compostela, Spain. [14]Department of Mathematics, Universitat Autònoma de Barcelona, Bellaterra, Spain. [279]These authors contributed equally: Luis A. Pérez-Jurado, Alejandro Cáceres. ✉e-mail: luis.perez@upf.edu; juanr.gonzalez@isglobal.org

## SCOURGE Cohort Group

Javier Abellán[15,16], René Acosta-Isaac[17], Jose María Aguado[18,19,20,21], Carlos Aguilar[22], Sergio Aguilera-Albesa[23,24], Abdolah Ahmadi Sabbagh[25], Jorge Alba[26], Sergiu Albu[27,28,29], Karla A. M. Alcalá-Gallardo[30], Julia Alcoba-Florez[31], Sergio Alcolea Batres[32], Holmes Rafael Algarin-Lara[33,34], Virginia Almadana[35], Kelliane A. Medeiros[36,37], Julia Almeida[38,39], Berta Almoguera[3,40], María R. Alonso[41], Nuria Álvarez[41], Rodolfo Álvarez-Sala Walther[32], Yady Álvarez-Benítez[33,34], Felipe Álvarez-Navia[42,43], Katiusse A. dos Santos[44], Álvaro Andreu-Bernabeu[20,45], Maria Rosa Antonijoan[46], Eleno Martínez-Aquino[47], Eunate Arana-Arri[48,49], Carlos Aranda[50,51], Celso Arango[20,45,52], Carolina Araque[53,54], Nathalia K. Araujo[55], Ana C. Arcanjo[56,57,58], Ana Arnaiz[59,60], Francisco Arnalich Fernández[61], María J. Arranz[62], José Ramón Arribas López[61], Maria-Jesús Artiga[63], Yubelly Avello-Malaver[64], Carmen Ayuso[3,40], Belén Ballina Martín[25], Raúl C. Baptista-Rosas[65,66,67], Ana María Baldion[64], Andrea Barranco-Díaz[34], María Barreda-Sánchez[68,69], Viviana Barrera-Penagos[64], Moncef Belhassen-Garcia[43,70], David Bernal-Bello[71], Enrique Bernal[68], Joao F. Bezerra[72], Marcos A. C. Bezerra[73], Natalia Blanca-López[74], Rafael Blancas[75], Lucía Boix-Palop[76], Alberto Borobia[77], Elsa Bravo[78], María Brion[79,80], Óscar Brochado-Kith[81], Ramón Brugada[80,82,83,84], Matilde Bustos[85], Alfonso Cabello[86], Alejandro Cáceres ![ORCID][4,5,279], Juan J. Cáceres-Agra[87], Esther Calbo[76], Enrique J. Calderón[6,88,89], Shirley Camacho[90], Francisco C. Ceballos[81], Yolanda Cañadas[51], Cristina Carbonell[42,43], Servando Cardona-Huerta[91], María Sánchez-Carpintero Abad[50,51], Carlos Carpio Segura[32], José Antonio Carrillo-Avila[92], Marcela C. Campos[56], Carlos Casasnovas[3,93,94], Luis Castaño[3,48,95,96,97], Carlos F. Castaño[50,51], Jose E. Castelao[98], Aranzazu Castellano Candalija[99], María A. Castillo[90], Walter G. Chaves-Santiago[54,100], Sylena Chiquillo-Gómez[33,34], Marco A. Cid-López[30], Óscar Cienfuegos-Jiménez[91], Rosa Conde-Vicente[101], Gabriela C. R. Cunha[102], M. Lourdes Cordero-Lorenzana[103], Dolores Corella[104,105], Almudena Corrales[106,107], Jose L. Cortés-Sánchez[91,108], Marta Corton[3,40], Karla S. C. Souza[109], Fabiola T. C. Silva[56], Raquel Cruz ![ORCID][3,8,9,10], Luisa Cuesta[110], Nathali A. C. Tavares[111], Maria C. C. Carvalho[112], David Dalmau[62,76], Raquel C. S. Dantas-Komatsu[113], M. Teresa Darnaude[114], Raimundo de Andrés[115], Carmen de Juan[116], Juan J. de la Cruz Troca[6,117,118], Carmen de la Horra[89], Ana B. de la Hoz[48], Alba De Martino-Rodríguez[119,120], Marina S. Cruz[121], Julianna Lys de Sousa Alves Neri[122], Victor del Campo-Pérez[123], Juan Delgado-Cuesta[124], Aranzazu Diaz de Bustamante[114], Anderson Díaz-Pérez[34], Beatriz Dietl[76], Silvia Diz-de Almeida[3,10], Manoella do Monte Alves[125,126], Elena Domínguez-Garrido[127], Lidia S. Rosa[128], Andre D. Luchessi[129], Jose Echave-Sustaeta[130], Rocío Eiros[131], César O. Enciso-Olivera[53,54], Gabriela Escudero[132], Pedro Pablo España[133], Gladys Estigarribia Sanabria[134],

**Article**

María Carmen Fariñas[59,60,135], Ramón Fernández[59,136], Lidia Fernández-Caballero[3,40], Ana Fernández-Cruz[137], Silvia Fernández-Ferrero[25], Yolanda Fernández Martínez[25], María J. Fernandez-Nestosa[138], Uxía Fernández-Robelo[139], Amanda Fernández-Rodríguez[81], Marta Fernández-Sampedro[59,60,135], Ruth Fernández[3,40], Tania Fernández-Villa[140], Carmen Fernández-Capitán[99], Antonio Augusto F. Carioca[141], Patricia Flores-Pérez[142], Lácides Fuenmayor-Hernández[34], Marta Fuertes-Núñez[25], Victoria Fumadó[143], Ignacio Gadea[144], Lidia Gagliardi[50,51], Manuela Gago-Domínguez[9,13], Natalia Gallego[11], Cristina Galoppo[145], Ana García-Soidán[146], Carlos García-Cerrada[15,16], Aitor García-de-Vicuña[48,95], Josefina Garcia-García[68], Irene García-García[77], Carmen García-Ibarbia[59,60,135], Andrés C. García-Montero[147], Leticia García[50,51], Mercedes García[50,51], María Carmen García Torrejón[16,148], Inés García[3,40], Elisa García-Vázquez[68], Emiliano Garza-Frias[91], Angela Gentile[145], Belén Gil-Fournier[149], Jéssica N. G. de Araújo[150], Mario Gómez-Duque[54,100], Javier Gómez-Arrue[119,120], Luis Gómez Carrera[32], María Gómez García[8], Ángela Gómez Sacristán[151], Juan R. González[4,5,6,14], Anna González-Neira[41], Beatriz González Álvarez[119,120], Fernán González Bernaldo de Quirós[152], Rafaela González-Montelongo[153], Javier González-Peñas[20,45,52], Manuel Gonzalez-Sagrado[101], Hugo Gonzalo-Benito[154], Oscar Gorgojo-Galindo[155], Miguel Górgolas[86], Florencia Guaragna[145], Jessica G. Chaux[54], Encarna Guillén-Navarro[68,156,157,158], Beatriz Guillén-Guío[106], Pablo Guisado-Vasco[130], Luz D. Gutiérrez-Castañeda[54,159], Juan F. Gutiérrez-Bautista[160], Sara Heili-Frades[161], Rafael H. Jacomo[162], Estefania Hernández[163], Cristina Hernández-Moro[25], Luis D. Hernández-Ortega[164,165], Guillermo Hernández-Pérez[42], Rebeca Hernández-Vaquero[166], Belén Herráez[41], M. Teresa Herranz[68], María Herrera[50,51], María José Herrero[167,168], Antonio Herrero-González[169], Juan P. Horcajada[28,170,171,172], Natale Imaz-Ayo[48], Maider Intxausti-Urrutibeaskoa[173], Antonio Íñigo-Campos[153], María Íñiguez[174], Rubén Jara[68], Ángel Jiménez[50,51], Ignacio Jiménez-Alfaro[175], Pilar Jiménez[160], María A. Jiménez-Sousa[81], Iolanda Jordan[6,176,177], Rocío Laguna-Goya[178,179], Daniel Laorden[32], María Lasa-Lázaro[178,179], María Claudia Lattig[90,180], Ailen Lauriente[145], Anabel Liger Borja[181], Lucía Llanos[182], Amparo López-Bernús[42,43], Miguel López de Heredia[3], Esther Lopez-Garcia[6,117,118,183], Eduardo López-Granados[3,184,185], Rosario Lopez-Rodriguez[3,40], Miguel A. López-Ruz[186,187,188], Leonardo Lorente[189], José M. Lorenzo-Salazar[153], José E. Lozano[190], María Lozano-Espinosa[181], Ignacio Mahillo[107,191,192], Esther Mancebo[178,179], Carmen Mar[133], Cristina Marcelo Calvo[99], Alba Marcos-Delgado[193], Miguel Marcos[42,43], Alicia Marín-Candón[77], Pablo Mariscal-Aguilar[32], Laura Martin-Pedraza[74], Marta Martin-Fernandez[194], Caridad Martín-López[181], José-Ángel Martín-Oterino[42,43], María Dolores Martín[195], Vicente Martín[6,193], María M. Martín[196], María Martín-Vicente[81], Amalia Martinez[197], Óscar Martínez-González[75], Ricardo Martínez[163], Pedro Martinez-Paz[154], Covadonga M. Díaz-Caneja[20,45,52], Óscar Martínez-Nieto[64,180], Iciar Martínez-López[198,199], Michel F. Martínez-Reséndez[91], Silvia Martínez[59,135], Juan José Martínez[3,94], Ángel Martínez-Pérez[200], Andrea Martínez-Ramas[3,40], Violeta Martínez-Robles[25], Laura Marzal[3,40], Juliana F. Mazzeu[201,202,203], Francisco J. Medrano[6,88,89], Xose M. Meijome[204,205], Natalia Mejuto-Montero[206], Ingrid Mendes[3], Alice L. Duarte[109], Ana Méndez-Echevarría[207], Humberto Mendoza Charris[34,78], Eleuterio Merayo Macías[208], Fátima Mercadillo[209], Arieh R. Mercado-Sesma[164,165], Pablo Mínguez[3,40], Elena Molina-Roldán[210], Antonio J. J. Molina[193], Juan José Montoya[163], Susana M. T. Pinho[36,211,212], Patricia Moreira-Escriche[116], Xenia Morelos-Arnedo[34,78], Rocío Moreno[3], Víctor Moreno Cuerda[15,16], Antonio Moreno-Docón[68], Junior Moreno-Escalante[34], Alberto Moreno Fernández[99], Patricia Muñoz García[20,107,213], Pablo Neira[145], Julián Nevado[3,11,12], Israel Nieto-Gañán[146], Vivian N. Silbiger[129], Rocío Nuñez-Torres[41], Antònia Obrador-Hevia[214,215], J. Gonzalo Ocejo-Vinyals[59,135], Virginia Olivar[145], Silviene F. Oliveira[56,203,216,217], Lorena Ondo[3,40], Alberto Orfao[38,39], Eva Ortega-Paino[63], Luis Ortega[218], Rocío Ortiz-López[91], Fernando Ortiz-Flores[59,135], José A. Oteo[26,174], Manuel Pacheco[163], Fredy Javier Pacheco-Miranda[34], Irene Padilla-Conejo[25], Sonia Panadero-Fajardo[92], Mara Parellada[20,45,52], Roberto Pariente-Rodríguez[146], Vicente Friaza[6,89], Estela Paz-Artal[178,179,219], Germán Peces-Barba[107,220], Miguel S. Pedromingo Kus[221], Celia Perales[144], Ney P. C. Santos[222], Genilson P. Guegel[223], María Jazmín Pérez[145], Alexandra Pérez[80,82], Patricia Pérez-Matute[174], César Pérez[224], Gustavo Pérez-de-Nanclares[48,95], Felipe Pérez-García[225,226], Patricia Pérez[227], Luis A. Pérez-Jurado ®[1,2,3,279] ✉, M. Elena Pérez-Tomás[68], Teresa Perucho[228], Lisbeth A. Pichardo[25], Adriana P. Ribeiro[36,37,212], Mel·lina Pinsach-Abuin[80,82], Luz Adriana Pinzón[54,100], Jeane F. P. Medeiros[229], Guillermo Pita[41], Francesc Pla-Juncà[3,230], Laura Planas-Serra[3,94], Ericka N. Pompa-Mera[231], Gloria L. Porras-Hurtado[163], Aurora Pujol[3,94,232], María Eugenia Quevedo-Chávez[33,34], Maria Angeles Quijada[46,233], Inés Quintela[8], Soraya Ramiro-León[149], Pedro Rascado Sedes[234], Joana F. R. Nunes[56], Delia Recalde[119,120], Emma Recio-Fernández[174], Salvador Resino[81], Renata R. Sousa[212], Carlos S. Rivadeneira-Chamorro[54], Diana Roa-Agudelo[64], Montserrat Robelo Pardo[234], Marianne R. Fernandes[222,235], María A. Rodríguez-Hernández[85], Agustí Rodriguez-Palmero[94,236], Emilio Rodríguez-Ruiz[9,234], Marilyn Johanna Rodriguez[54], Fernando Rodríguez-Artalejo[6,117,118,183], Marena Rodríguez-Ferrer[34], Carlos Rodríguez-Gallego[237,238], José A. Rodríguez-García[25], Belén Rodríguez Maya[15], Antonio Rodriguez-Nicolas[160], German Ezequiel Rodríguez-Novoa[145], Paula A. Rodriguez-Urrego[64], Federico Rojo[239,240], Andrea Romero-Coronado[34], Rubén Morilla[89,241], Filomeno Rondón-García[25], Antonio Rosales-Castillo[242], Cladelis Rubio[243], María Rubio Olivera[50,51], Francisco Ruiz-Cabello[160,187,244], Eva Ruiz-Casares[228], Juan J. Ruiz-Cubillan[59,135], Javier Ruiz-Hornillos[51,245,246], Montserrat Ruiz[3,94], Pablo Ryan[247,248,249], Hector D. Salamanca[53,54], Lorena Salazar-García[90], Giorgina Gabriela Salgueiro-Origlia[99], Anna Sangil[76], Olga Sánchez-Pernaute[250], Pedro-Luis Sánchez[43,131], Antonio J. Sánchez López[251], Clara Sánchez-Pablo[131],

María Concepción Sánchez-Prados[32], Javier Sánchez-Real[25], Jorge Sánchez-Redondo[15,252], Cristina Sancho-Sainz[173], Esther Sande[224], Arnoldo Santos[224], Agatha Schlüter[3,94], Sonia Segovia[230,253,254], Alex Serra-Llovich[62], Fernando Sevil-Puras[22], Marta Sevilla-Porras[3,11], Miguel A. Sicolo[255,256], Cristina Silván-Fuentes[3], Vitor M. S. Moraes[257], Vanessa S. Souza[102], Jordi Solé-Violán[107,258], José Manuel Soria[200], Jose V. Sorlí[104,105], Nayara S. Silva[259], Juan Carlos Souto[17], John J. Sprockel[54,100], José Javier Suárez-Rama[8], David A. Suárez-Zamora[64], Xiana Taboada-Fraga[206], Eduardo Tamayo[155,260], Alvaro Tamayo-Velasco[261], Juan Carlos Taracido-Fernández[169], Romero H. T. Vasconcelos[111], Carlos Tellería[119,120], Thássia M. T. Carratto[257], Jair Antonio Tenorio-Castaño[3,11,12], Alejandro Teper[145], Izabel M. T. Araujo[109], Juan Torres-Macho[262], Lilian Torres-Tobar[263], Ronald P. Torres-Gutiérrez[221], Jesús Troya[247], Miguel Urioste[209], Juan Valencia-Ramos[264], Agustín Valido[35,265], Juan Pablo Vargas-Gallo[266,267], Belén Varón[268], Tomas Vega[269], Santiago Velasco-Quirce[270], Valentina Vélez-Santamaría[93,94], Virginia Víctor[50,51], Julia Vidán-Estévez[25], Gabriela V. Silva[109], Miriam Vieitez-Santiago[59,135], Carlos Vilches[271], Lavinia Villalobos[25], Felipe Villar[220], Judit Villar-Garcia[272,273,274], Cristina Villaverde[3,40], Pablo Villoslada-Blanco[174], Ana Virseda-Berdices[81], Tatiana X. Costa[275], Zuleima Yáñez[34], Antonio Zapatero-Gaviria[276], Ruth Zarate[277], Sandra Zazo[239], Carlos Flores[106,107,153,238], José A. Riancho[59,60,135], Augusto Rojas-Martinez[278], Pablo Lapunzina [ORCID][3,11,12] & Ángel Carracedo[3,8,9,10,13]

[15]Hospital Universitario Mostoles, Medicina Interna, Madrid, Spain. [16]Universidad Francisco de Vitoria, Madrid, Spain. [17]Haemostasis and Thrombosis Unit, Hospital de la Santa Creu I Sant Pau, IIB Sant Pau, Barcelona, Spain. [18]Unit of Infectious Diseases, Hospital Universitario 12 de Octubre, Instituto de Investigación Sanitaria Hospital 12 de Octubre (imas12), Madrid, Spain. [19]Spanish Network for Research in Infectious Diseases (REIPI RD16/0016/0002), Instituto de Salud Carlos III, Madrid, Spain. [20]School of Medicine, Universidad Complutense, Madrid, Spain. [21]Centre for Biomedical Network Research on Infectious Diseases, Instituto de Salud Carlos III, Madrid, Spain. [22]Hospital General Santa Bárbara de Soria, Soria, Spain. [23]Pediatric Neurology Unit, Department of Pediatrics, Navarra Health Service Hospital, Pamplona, Spain. [24]Navarra Health Service, NavarraBioMed Research Group, Pamplona, Spain. [25]Complejo Asistencial Universitario de León, León, Spain. [26]Hospital Universitario San Pedro, Infectious Diseases Department, Logroño, Spain. [27]Fundación Institut Guttmann, Institut Universitari de Neurorehabilitació ofounde a la UAB, Hospital de Neurorehabilitació, Barcelona, Spain. [28]Universitat Autònoma de Barcelona (UAB), Barcelona, Spain. [29]Fundació Institut d'Investigació en Ciències de la Salut Germans Trias I Pujol, Barcelona, Spain. [30]Hospital General de Occidente, Guadalajara, Mexico. [31]Microbiology Unit, Hospital Universitario N. S. de Candelaria, Santa Cruz de Tenerife, Spain. [32]Hospital Universitario La Paz-IDIPAZ, Servicio de Neumología, Madrid, Spain. [33]Camino Universitario Adelita de Char, Mired IPS, Barranquilla, Colombia. [34]Universidad Simón Bolívar, Facultad de Ciencias de la Salud, Barranquilla, Colombia. [35]Hospital Universitario Virgen Macarena, Neumología, Seville, Spain. [36]Hospital das Forças Armadas, Brasília, Brazil. [37]Exército Brasileiro, Brasília, Brazil. [38]Departamento de Medicina, Universidad de Salamanca, Salamanca, Spain. [39]Centro de Investigación del Cáncer (IBMCC) Universidad de Salamanca – CSIC, Salamanca, Spain. [40]Department of Genetics & Genomics, Instituto de Investigación Sanitaria-Fundación Jiménez Díaz University Hospital – Universidad Autónoma de Madrid (IIS-FJD, UAM), Madrid, Spain. [41]Spanish National Cancer Research Centre, Human Genotyping-CEGEN Unit, Madrid, Spain. [42]Hospital Universitario de Salamanca-IBSAL, Servicio de Medicina Interna, Salamanca, Spain. [43]Universidad de Salamanca, Salamanca, Spain. [44]Universidade Federal do Rio Grande do Norte, Programa de Pós-Graduação em Ciências Farmacêuticas, Natal, Brazil. [45]Department of Child and Adolescent Psychiatry, Institute of Psychiatry and Mental Health, Hospital General Universitario Gregorio Marañón (IiSGM), Madrid, Spain. [46]Clinical Pharmacology Service, Hospital de la Santa Creu I Sant Pau, IIB Sant Pau, Barcelona, Spain. [47]Servicio de Medicina Interna, Sanatorio Franchin, Buenos Aires, Argentina. [48]Biocruces Bizkai HRI, Bizkaia, Spain. [49]Cruces University Hospital, Osakidetza, Bizkaia, Spain. [50]Hospital Infanta Elena, Valdemoro, Madrid, Spain. [51]Instituto de Investigación Sanitaria-Fundación Jiménez Díaz University Hospital – Universidad Autónoma de Madrid (IIS-FJD, UAM), Madrid, Spain. [52]Centre for Biomedical Network Research on Mental Health (CIBERSAM), Instituto de Salud Carlos III, Madrid, Spain. [53]Fundación Hospital Infantil Universitario de San José, Bogotá, Colombia. [54]Fundación Universitaria de Ciencias de la Salud, Bogotá, Colombia. [55]Universidade Federal do Rio Grande do Norte, Departamento de Analises Clínicas e Toxicológicas, Natal, Brazil. [56]Departamento de Genética e Morfologia, Instituto de Ciências Biológicas, Universidade de Brasília, Brasília, Brazil. [57]Colégio Marista de Brasilia, Brasília, Brazil. [58]Associação Brasileira de Educação e Cultura, Ribeirão Preto, Brazil. [59]IDIVAL, Santander, Spain. [60]Universidad de Cantabria, Santander, Spain. [61]Hospital Universitario La Paz-IDIPAZ, Servicio de Medicina Interna, Madrid, Spain. [62]Fundació Docència I Recerca Mutua Terrassa, Barcelona, Spain. [63]Spanish National Cancer Research Center, CNIO Biobank, Madrid, Spain. [64]Fundación Santa Fe de Bogota, Departamento Patologia y Laboratorios, Bogotá, Colombia. [65]Hospital General de Occidente, Zapopan, Jalisco, Mexico. [66]Centro Universitario de Tonalá, Universidad de Guadalajara, Tonalá, Jalisco, Mexico. [67]Centro de Investigación Multidisciplinario en Salud, Universidad de Guadalajara, Tonalá, Jalisco, Mexico. [68]Instituto Murciano de Investigación Biosanitaria (IMIB-Arrixaca), Murcia, Spain. [69]Universidad Católica San Antonio de Murcia (UCAM), Murcia, Spain. [70]Hospital Universitario de Salamanca-IBSAL, Servicio de Medicina Interna-Unidad de Enfermedades Infecciosas, Salamanca, Spain. [71]Hospital Universitario de Fuenlabrada, Department of Internal Medicine, Madrid, Spain. [72]Escola Tecnica de Saúde, Laboratorio de Vigilancia Molecular Aplicada, Recife, Brazil. [73]Federal University of Pernambuco, Genetics Postgraduate Program, Recife, Pernambuco, Brazil. [74]Hospital Universitario Infanta Leonor, Servicio de Alergia, Madrid, Spain. [75]Hospital Universitario del Tajo, Servicio de Medicina Intensiva, Toledo, Spain. [76]Hospital Universitario Mutua Terrassa, Barcelona, Spain. [77]Hospital Universitario La Paz-IDIPAZ, Servicio de Farmacología, Madrid, Spain. [78]Alcaldía de Barranquilla, Secretaría de Salud, Barranquilla, Colombia. [79]Instituto de Investigación Sanitaria de Santiago (IDIS), Xenética Cardiovascular, Santiago de Compostela, Spain. [80]Centre for Biomedical Network Research on Cardiovascular Diseases (CIBERCV), Instituto de Salud Carlos III, Madrid, Spain. [81]Unidad de Infección Viral e Inmunidad, Centro Nacional de Microbiología (CNM), Instituto de Salud Carlos III (ISCIII), Madrid, Spain. [82]Cardiovascular Genetics Center, Institut d'Investigació Biomèdica Girona (IDIBGI), Girona, Spain. [83]Medical Science Department, School of Medicine, University of Girona, Girona, Spain. [84]Hospital Josep Trueta, Cardiology Service, Girona, Spain. [85]Institute of Biomedicine of Seville (IbiS), Consejo Superior de Investigaciones Científicas (CSIC)- University of Seville- Virgen del Rocio University Hospital, Seville, Spain. [86]Division of Infectious Diseases, Instituto de Investigación Sanitaria-Fundación Jiménez Díaz University Hospital – Universidad Autónoma de Madrid (IIS-FJD, UAM), Madrid, Spain. [87]Intensive Care Unit, Hospital Universitario Insular de Gran Canaria, Las Palmas de Gran Canaria, Spain. [88]Departemente de Medicina, Hospital Universitario Virgen del Rocío, Universidad de Sevilla, Seville, Spain. [89]Instituto de Biomedicina de Sevilla, Seville, Spain. [90]Universidad de los Andes, Facultad de Ciencias, Bogotá, Colombia. [91]Tecnológico de Monterrey, Monterrey, Mexico. [92]Andalusian Public Health System Biobank, Granada, Spain. [93]Neuromuscular Unit, Neurology Department, Hospital Universitari de Bellvitge, L'Hospitalet de Llobregat, Spain. [94]Bellvitge Biomedical Research Institute (IDIBELL), Neurometabolic Diseases Laboratory, L'Hospitalet de Llobregat, Spain. [95]Osakidetza, Cruces University Hospital, Barakaldo, Spain. [96]Centre for Biomedical Network Research on Diabetes and Metabolic Associated Diseases (CIBERDEM), Instituto de Salud Carlos III, Madrid, Spain. [97]University of Pais Vasco, UPV/EHU, Bilbao, Spain. [98]Oncology and Genetics Unit, Instituto de Investigacion Sanitaria Galicia Sur, Xerencia de Xestion Integrada de Vigo-Servizo Galego de Saúde, Vigo, Spain. [99]Hospital Universitario La Paz, Hospital Carlos III, Madrid, Spain. [100]Hospital de San José, Sociedad de Cirugía de Bogota, Bogotá, Colombia. [101]Hospital Universitario Río Hortega, Valladolid, Spain. [102]Programa de Pós Graduação em Ciências da Saúde, Faculdade de Medicina,

Universidade de Brasília, Brasília, Brazil. [103]Servicio de Medicina ofounded, Complejo Hospitalario Universitario de A Coruña (CHUAC), Sistema Galego de Saúde (SERGAS), A Coruña, Spain. [104]Valencia University, Preventive Medicine Department, Valencia, Spain. [105]Centre for Biomedical Network Research on Physiopatology of Obesity and Nutrition (CIBEROBN), Instituto de Salud Carlos III, Madrid, Spain. [106]Research Unit, Hospital Universitario N.S. de Candelaria, Santa Cruz de Tenerife, Spain. [107]Centre for Biomedical Network Research on Respiratory Diseases (CIBERES), Instituto de Salud Carlos III, Madrid, Spain. [108]Otto von Guericke University, Departament of Microgravity and Translational Regenerative Medicine, Magdeburg, Germany. [109]Universidade Federal do Rio Grande do Norte, Departamento de Analises Clinicais e Toxicologias, Natal, Brazil. [110]Institute of Psychiatry and Mental Health, Hospital General Universitario Gregorio Marañón (IiSGM), Madrid, Spain. [111]Hospital Universitario Lauro Wanderley, João Pessoa, Brazil. [112]Programa de Pós Graduação em Ciências Farmacêuticas (PPgCF), Natal, Brazil. [113]Universidade Federal do Rio Grande do Norte, Programa de Pós-graduação em Ciências da Saúde, Natal, Brazil. [114]Hospital Universitario Mostoles, Unidad de Genética, Madrid, Spain. [115]Internal Medicine Department, Instituto de Investigación Sanitaria-Fundación Jiménez Díaz University Hospital – Universidad Autónoma de Madrid (IIS-FJD, UAM), Madrid, Spain. [116]Hospital Universitario Severo Ochoa, Servicio de Medicina Interna, Madrid, Spain. [117]Department of Preventive Medicine and Public Health, School of Medicine, Universidad Autónoma de Madrid, Madrid, Spain. [118]IdiPaz (Instituto de Investigación Sanitaria Hospital Universitario La Paz), Madrid, Spain. [119]Instituto Aragonés de Ciencias de la Salud (IACS), Zaragoza, Spain. [120]Instituto Investigación Sanitaria Aragón (IIS-Aragon), Zaragoza, Spain. [121]Universidade Federal do Rio Grande do Norte, Programa de Pós-Graduação em Ciências da Saúde, Natal, Brazil. [122]Universidade Federal do Rio Grande do Norte, Programa de Pós Graduação em Nutrição, Natal, Brazil. [123]Preventive Medicine Department, Instituto de Investigacion Sanitaria Galicia Sur, Xerencia de Xestion Integrada de Vigo-Servizo Galego de Saúde, Vigo, Spain. [124]Hospital Universitario Virgen del Rocío, Servicio de Medicina Interna, Seville, Spain. [125]Universidade Federal do Rio Grande do Norte, Departamento de Infectologia, Natal, Brazil. [126]Hospital de Doenças Infecciosas Giselda Trigueiro, Rio Grande do Norte, Natal, Brazil. [127]Unidad Diagnóstico Molecular. Fundación Rioja Salud, Logroño, Spain. [128]Faculdade de Ciências da Saúde, Universidade de Brasília, Brasília, Brazil. [129]Universidade Federal do Rio Grande do Norte, Departamento de Analises Clinicas e Toxicologicas, Natal, Brazil. [130]Hospital Universitario Quironsalud Madrid, Madrid, Spain. [131]Hospital Universitario de Salamanca-IBSAL, Servicio de Cardiología, Salamanca, Spain. [132]Hospital Universitario Puerta de Hierro, Servicio de Medicina Interna, Majadahonda, Spain. [133]Biocruces Bizkaia Health Research Institute, Galdakao University Hospital, Osakidetza, Barakaldo, Spain. [134]Instituto Regional de Investigación en Salud-Universidad Nacional de Caaguazú, Caaguazú, Paraguay. [135]Hospital U M Valdecilla, Santander, Spain. [136]Fundación Asilo San Jose, Santander, Spain. [137]Unidad de Enfermedades Infecciosas, Servicio de Medicina Interna, Hospital Universitario Puerta de Hierro, Instituto de Investigación Sanitaria Puerta de Hierro – Segovia de Arana, Madrid, Spain. [138]Universidad Nacional de Asunción, Facultad de Politécnica, San Lorenzo, Paraguay. [139]Urgencias Hospitalarias, Complejo Hospitalario Universitario de A Coruña (CHUAC), Sistema Galego de Saúde (SERGAS), A Coruña, Spain. [140]Grupo de Investigación en Interacciones Gen-Ambiente y Salud (GIIGAS) – Instituto de Biomedicina (IBIOMED), Universidade de León, León, Spain. [141]Universidade de Fortaleza, Natal, Brazil. [142]Hospital Universitario Niño Jesús, Pediatrics Department, Madrid, Spain. [143]Unitat de Malalties Infeccioses I Importades, Servei de Pediatría, Infectious and Imported Diseases, Pediatric Unit, Hospital Universitari Sant Joan de Deú, Barcelona, Spain. [144]Microbiology Department, Instituto de Investigación Sanitaria-Fundación Jiménez Díaz University Hospital – Universidad Autónoma de Madrid (IIS-FJD, UAM), Madrid, Spain. [145]Hospital de Niños Ricardo Gutierrez, Buenos Aires, Argentina. [146]Department of Immunology, IRYCIS, Hospital Universitario Ramón y Cajal, Madrid, Spain. [147]University of Salamanca, Biomedical Research Institute of Salamanca (IBSAL), Salamanca, Spain. [148]Hospital Infanta Elena, Servicio de Medicina Intensiva, Valdemoro, Madrid, Spain. [149]Hospital Universitario de Getafe, Unidad de Genética, Madrid, Spain. [150]Programa de pós-graduação em biotecnologia – Rede Nordeste de Biotecnologia (RENORBIO), Universidade Federal do Rio Grande do Norte, Natal, Brazil. [151]Pneumology Department, Hospital General Universitario Gregorio Marañón (iiSGM), Madrid, Spain. [152]Ministerio de Salud Ciudad de Buenos Aires, Buenos Aires, Argentina. [153]Genomics Division, Instituto Tecnológico y de Energías Renovables, Santa Cruz de Tenerife, Spain. [154]Hospital Clinico Universitario de Valladolid, Unidad de Apoyo a la Investigación, Valladolid, Spain. [155]Universidad de Valladolid, Departamento de Cirugía, Valladolid, Spain. [156]Sección Genética Médica – Servicio de Pediatría, Hospital Clínico Universitario Virgen de la Arrixaca, Servicio Murciano de Salud, Murcia, Spain. [157]Departamento Cirugía, Pediatría, Obstetricia y Ginecología, Facultad de Medicina, Universidad de Murcia (UMU), Murcia, Spain. [158]Grupo Clínico Vinculado, Centre for Biomedical Network Research on Rare Diseases (CIBERER), Instituto de Salud Carlos III, Madrid, Spain. [159]Hospital Universitario Centro Dermatológico Federico Lleras Acosta, Bogotá, Colombia. [160]Hospital Universitario Virgen de las Nieves, Servicio de Análisis Clínicos e Inmunología, Granada, Spain. [161]Intermediate Respiratory Care Unit, Department of Neumology, Instituto de Investigación Sanitaria-Fundación Jiménez Díaz University Hospital – Universidad Autónoma de Madrid (IIS-FJD, UAM), Madrid, Spain. [162]Sabin Medicina Diagnóstica, Santa Catarina, Brazil. [163]Clinica Comfamiliar Risaralda, Pereira, Colombia. [164]Centro Universitario de Tonalá, Universidad de Guadalajara, Guadalajara, Mexico. [165]Centro de Investigación Multidisciplinario en Salud, Universidad de Guadalajara, Guadalajara, Mexico. [166]Unidad de Cuidados, Intensivos Hospital Clínico Universitario de Santiago (CHUS), Sistema Galego de Saúde (SERGAS), Santiago de Compostela, Spain. [167]IIS La Fe, Plataforma de Farmacogenética, Valencia, Spain. [168]Universidad de Valencia, Departamento de Farmacología, Valencia, Spain. [169]Data Analysis Department, Instituto de Investigación Sanitaria-Fundación Jiménez Díaz University Hospital – Universidad Autónoma de Madrid (IIS-FJD, UAM), Madrid, Spain. [170]Hospital del Mar, Infectious Diseases Service, Barcelona, Spain. [171]Institut Hospital del Mar d'Investigacions Mèdiques (IMIM), Barcelona, Spain. [172]CEXS-Universitat Pompeu Fabra, Spanish Network for Research in Infectious Diseases (REIPI), Barcelona, Spain. [173]Biocruces Bizkaia Health Research Institute, Basurto University Hospital, Osakidetza, Basurto, Spain. [174]Infectious Diseases, Microbiota and Metabolism Unit, Center for Biomedical Research of La Rioja (CIBIR), Logroño, Spain. [175]Opthalmology Department, Instituto de Investigación Sanitaria-Fundación Jiménez Díaz University Hospital – Universidad Autónoma de Madrid (IIS-FJD, UAM), Madrid, Spain. [176]Hospital Sant Joan de Deu,Pediatric Critical Care Unit, Barcelona, Spain. [177]Paediatric Intensive Care Unit, Agrupación Hospitalaria Clínic-Sant Joan de Déu, Esplugues de Llobregat, Barcelona, Spain. [178]Hospital Universitario 12 de Octubre, Department of Immunology, Madrid, Spain. [179]Instituto de Investigación Sanitaria Hospital 12 de Octubre (imas12), Transplant Immunology and Immunodeficiencies Group, Madrid, Spain. [180]SIGEN Alianza Universidad de los Andes – Fundación Santa Fe de Bogotá, Bogotá, Colombia. [181]Hospital General de Segovia, Medicina Intensiva, Segovia, Spain. [182]Clinical Trials Unit, Instituto de Investigación Sanitaria-Fundación Jiménez Díaz University Hospital – Universidad Autónoma de Madrid (IIS-FJD, UAM), Madrid, Spain. [183]IMDEA-Food Institute, CEI UAM + CSIC, Madrid, Spain. [184]Hospital Universitario La Paz-IDIPAZ, Servicio de Inmunología, Madrid, Spain. [185]La Paz Institute for Health Research (IdiPAZ), Lymphocyte Pathophysiology in Immunodeficiencies Group, Madrid, Spain. [186]Hospital Universitario Virgen de las Nieves, Servicio de Enfermedades Infecciosas, Granada, Spain. [187]Instituto de Investigación Biosanitaria de Granada (ibs GRANADA), Granada, Spain. [188]Universidad de Granada, Departamento de Medicina, Granada, Spain. [189]Intensive Care Unit, Hospital Universitario de Canarias, La Laguna, Spain. [190]Dirección General de Salud Pública, Consejería de Sanidad, Junta de Castilla y León, Valladolid, Spain. [191]Fundación Jiménez Díaz, Epidemiology, Madrid, Spain. [192]Universidad Autónoma de Madrid, Department of Medicine, Madrid, Spain. [193]Instituto de Biomedicina (IBIOMED), Universidad de León, León, Spain. [194]Universidad de Valladolid, Departamento de Medicina, Valladolid, Spain. [195]Preventive Medicine Department, Instituto de Investigación Sanitaria-Fundación Jiménez Díaz University Hospital – Universidad Autónoma de Madrid (IIS-FJD, UAM), Madrid, Spain. [196]Intensive Care Unit, Hospital Universitario N. S. de Candelaria, Santa Cruz de Tenerife, Spain. [197]Hospital Universitario Infanta Leonor, Servicio de Medicina Intensiva, Madrid, Spain. [198]Unidad de Genética y Genómica Islas Baleares, Palma de Mallorca, Spain. [199]Hospital Universitario Son Espases, Unidad de Diagnóstico Molecular y Genética Clínica, Palma de Mallorca, Spain. [200]Genomics of Complex Diseases Unit, Research Institute of Hospital de la Santa Creu I Sant Pau, IIB Sant Pau, Barcelona, Spain. [201]Faculdade de Medicina, Universidade de Brasília, Brasília, Brazil. [202]Programa de Pós-Graduação em Ciências Médicas, Universidade de Brasília, Brasília, Brazil. [203]Programa de Pós-Graduação em Ciências da Saúde, Universidade de Brasília, Brasília, Brazil. [204]Hospital El Bierzo, Gerencia de Asistencia Sanitaria del Bierzo (GASBI), Gerencia Regional de Salud (SACYL),

Ponferrada, Spain. [205]Grupo INVESTEN, Instituto de Salud Carlos III, Madrid, Spain. [206]Unidad de Cuidados Intensivos, Complejo Universitario de A Coruña (CHUAC), Sistema Galego de Saúde (SERGAS), A Coruña, Spain. [207]Hospital Universitario La Paz-IDIPAZ, Servicio de Pediatría, Madrid, Spain. [208]Hospital El Bierzo, Unidad Cuidados Intensivos, León, Spain. [209]Spanish National Cancer Research Centre, Familial Cancer Clinical Unit, Madrid, Spain. [210]Instituto de Investigación Sanitaria San Carlos (IdISSC), Hospital Clínico San Carlos (HCSC), Madrid, Spain. [211]Marinha do Brasil, Brasil, Brazil. [212]Universidade de Brasília, Brasília, Brazil. [213]Hospital General Universitario Gregorio Marañón (IiSGM), Madrid, Spain. [214]Unidad de Genética y Genómica Islas Baleares,Unidad de Diagnóstico Molecular y Genética Clínica, Hospital Universitario Son Espases, Palma de Mallorca, Spain. [215]Instituto de Investigación Sanitaria Islas Baleares (IdISBa), Palma de Mallorca, Spain. [216]Programa de Pós-Graduação em Biologia Animal (UnB), Brasília, Brazil. [217]Programa de Pós-Graduação Profissional em Ensino de Biologia (UnB), Brasília, Brazil. [218]Anatomía Patológica, Instituto de Investigación Sanitaria San Carlos (IdISSC), Hospital Clínico San Carlos (HCSC), Madrid, Spain. [219]Universidad Complutense de Madrid, Department of Immunology, Ophthalmology and ENT, Madrid, Spain. [220]Department of Neumology, Instituto de Investigación Sanitaria-Fundación Jiménez Díaz University Hospital – Universidad Autónoma de Madrid (IIS-FJD, UAM), Madrid, Spain. [221]Hospital Nuestra Señora de Sonsoles, Ávila, Spain. [222]Universidade Federal do Pará, Núcleo de Pesquisas em Oncologia, Belém, Pará, Brazil. [223]Secretaria Municipal de Saude de Apodi, Natal, Brazil. [224]Intensive Care Department, Instituto de Investigación Sanitaria-Fundación Jiménez Díaz University Hospital – Universidad Autónoma de Madrid (IIS-FJD, UAM), Madrid, Spain. [225]Hospital Universitario Príncipe de Asturias, Servicio de Microbiología Clínica, Madrid, Spain. [226]Universidad de Alcalá de Henares, Departamento de Biomedicina y Biotecnología, Facultad de Medicina y Ciencias de la Salud, Madrid, Spain. [227]Inditex, A Coruña, Spain. [228]GENYCA, Madrid, Spain. [229]Universidade Federal do Rio Grande do Norte, Departamento de Análises Clínicas e Toxicológicas, Natal, Brazil. [230]Neuromuscular Diseases Unit, Department of Neurology, Hospital de la Santa Creu I Sant Pau, Universitat Autònoma de Barcelona, Barcelona, Spain. [231]Instituto Mexicano del Seguro Social (IMSS), Centro Médico Nacional Siglo XXI, Unidad de Investigación Médica en Enfermedades Infecciosas y Parasitarias, Mexico City, Mexico. [232]Catalan Institution of Research and Advanced Studies (ICREA), Barcelona, Spain. [233]Drug Research Centre, Institut d'Investigació Biomèdica Sant Pau, IIB-Sant Pau, Barcelona, Spain. [234]Unidad de Cuidados Intensivos, Hospital Clínico Universitario de Santiago (CHUS), Sistema Galego de Saúde (SERGAS), Santiago de Compostela, Spain. [235]Hospital Ophir Loyola, Departamento de Ensino e Pesquisa, Belém, Pará, Brazil. [236]University Hospital Germans Trias I Pujol, Pediatrics Department, Badalona, Spain. [237]Department of Immunology, Hospital Universitario de Gran Canaria Dr. Negrín, Las Palmas de Gran Canaria, Spain. [238]Department of Clinical Sciences, University Fernando Pessoa Canarias, Las Palmas de Gran Canaria, Spain. [239]Department of Pathology, Biobank, Instituto de Investigación Sanitaria-Fundación Jiménez Díaz University Hospital – Universidad Autónoma de Madrid (IIS-FJD, UAM), Madrid, Spain. [240]Centre for Biomedical Network Research on Cancer (CIBERONC), Instituto de Salud Carlos III, Madrid, Spain. [241]Universidad de Sevilla, Departamento de Enfermería, Seville, Spain. [242]Hospital Universitario Virgen de las Nieves, Servicio de Medicina Interna, Granada, Spain. [243]Fundación Universitaria de Ciencias de la Salud, Grupo de Ciencias Básicas en Salud (CBS), Bogotá, Colombia. [244]Universidad de Granada, Departamento Bioquímica, Biología Molecular e Inmunología III, Granada, Spain. [245]Hospital Infanta Elena, Allergy Unit, Valdemoro, Madrid, Spain. [246]Faculty of Medicine, Universidad Francisco de Vitoria, Madrid, Spain. [247]Hospital Universitario Infanta Leonor, Madrid, Spain. [248]Complutense University of Madrid, Madrid, Spain. [249]Gregorio Marañón Health Research Institute (IiSGM), Madrid, Spain. [250]Reumathology Service, Instituto de Investigación Sanitaria-Fundación Jiménez Díaz University Hospital – Universidad Autónoma de Madrid (IIS-FJD, UAM), Madrid, Spain. [251]Biobank, Puerta de Hierro-Segovia de Arana Health Research Institute, Madrid, Spain. [252]Universidad Rey Juan Carlos, Madrid, Spain. [253]The John Walton Muscular Dystrophy Research Centre, Newcastle University and Newcastle Hospitals NHS Foundation Trust, Newcastle upon Tyne, England. [254]Neuromuscular Unit, Neuro-pediatrics Department, Institut de Recerca Sant Joan de Déu, Hospital Sant Joan de Déu, Barcelona, Spain. [255]Casa de Saúde São Lucas, Natal, Brazil. [256]Hospital Rio Grande, Natal, Brazil. [257]Departamento de Química, Faculdade de Filosofia, Ciências e Letras de Ribeirão Preto, Universidade de São Paulo, São Paulo, Brazil. [258]Intensive Care Unit, Hospital Universitario de Gran Canaria Dr. Negrín, Las Palmas de Gran Canaria, Spain. [259]Universidade Federal do Rio Grande do Norte, Pós-graduação em Biotecnologia – Rede de Biotecnologia do Nordeste (Renorbio), Natal, Brazil. [260]Hospital Clinico Universitario de Valladolid, Servicio de Anestesiologia y Reanimación, Valladolid, Spain. [261]Hospital Clinico Universitario de Valladolid, Servicio de Hematologia y Hemoterapia, Valladolid, Spain. [262]Hospital Universitario Infanta Leonor, Servicio de Medicina Interna, Madrid, Spain. [263]Sociedad de Cirugía de Bogotá, Hospital de San José, Bogotá, Colombia. [264]University Hospital of Burgos, Burgos, Spain. [265]Universidad de Sevilla, Seville, Spain. [266]Fundación Santa Fe de Bogota, Instituto de servicios medicos de Emergencia y trauma, Bogotá, Colombia. [267]Universidad de los Andes, Bogotá, Colombia. [268]Quironprevención, A Coruña, Spain. [269]Junta de Castilla y León, Consejería de Sanidad, Valladolid, Spain. [270]Gerencia Atención Primaria de Burgos, Burgos, Spain. [271]Immunogenetics-Histocompatibility group, Servicio de Inmunología, Instituto de Investigación Sanitaria Puerta de Hierro – Segovia de Arana, Madrid, Spain. [272]Hospital del Mar, Department of Infectious Diseases, Barcelona, Spain. [273]Hospital del Mar Medical Research Institute (IMIM), Barcelona, Spain. [274]Universitat Autònoma de Barcelona, Department of Medicine, Barcelona, Spain. [275]Maternidade Escola Janário Cicco, Natal, Brazil. [276]Consejería de Sanidad, Comunidad de Madrid, Madrid, Spain. [277]Centro para el Desarrollo de la Investigación Científica, Caaguazú, Paraguay. [278]Tecnológico de Monterrey, Escuela de Medicina y Ciencias de la Salud, Monterrey, Mexico.

