## [Peer Review File · Communications Biology]

Reviewers' comments:

Reviewer #1 (Remarks to the Author):

1. Fig 1: the authors may wish to separate the data of sex-biased hospitalization & mortality rate and estimated prevalence of CME and LOY in men into different figures as the latter is only from the male population. Or a gender-specific style of combined figure is recommended.
2. A table of demographic parameters comparing control and LOY patients is needed.
3. It is surprising that the authors analyzed cell-type composition and functionality by whole blood RNA transcriptomic analysis using, which is a low-resolution research method and might cause over-interpretation of the data, especially the sample size is relatively small (LOY n=11, Control n=32). i.e., it is hard to imagine that the increased counts of endothelial cells were suggested without any other direct quantitative method. Further, transcriptomic and functional analyses of sorted blood cells is recommended for demonstrating the mechanistic basis.
4. Fig 3: Although the prevalence of LOY in males was considerably higher compared to the prevalence of detectable CME in the general population, the 90-day risk of COVID19 lethality was higher in CME than LOY (OR 1.78 vs. 1.40). Therefore, listing the contribution of gender-specific CMEs to COVID 19 lethality would provide a direct comparison between the contributions of male CME and LOY to COVID19 lethality.

Reviewer #2 (Remarks to the Author):

The paper from Perez-Jurado et al. carried out a LOY analysis in a large cohort of 9578 patients with COVID from SCOURGE study. They revealed three important findings: 1) the presence of CME in both sexes was significantly associated with COVID-19 lethality; 2) the presence of LOY in male was significantly associated with COVID-19 lethality; 3) LOY was associated with dysregulated transcriptomic of immune function that involved in response SARS-CoV2 infection. Overall, this study is very interesting because it provides evidence that host genetics underlie at least part of the sex-biased severity and mortality of COVID-19. I am supportive of its publication and I have several comments to this paper as follows.

1. I think the author should keep the use of terms and abbreviations along with the previous studies. For example, the authors use chromosomal mosaic events (CME) to refer autosomal and X alterations. However, the use of mosaic chromosomal alterations (mCAs) has spread in the field (e.g., PMID: 29995854; 32581363; 32581364; 34099924; 33866012). So, I suggest the authors to use mCAs and rephrase throughout.
2. In addition, the terms X chromosome monosomy (XCM) and LOY are overlapped. To avoid confusion, I suggest the authors to use LOY alone, because LOY can mean XCM in males.
3. Age is the biggest risk factor for COVID-19 in both males and females. As the authors have shown, individuals with CME have an average age (79.5 yrs) much higher than non-CME controls (70.6 yrs) and this is also the case for LOY male (83.5 yrs) and non-LOY males (69.1). So, I think the results that the presence of CME and LOY are significantly associated with COVID-19 lethality may be confounded by age and age should be adjusted.
4. Since CME has a higher OR value than LOY for 90-day COVID-19 lethality, it is interesting to consider sex in the analysis of the association of CME with 90-day COVID-19 lethality.

5. I noted that the prevalence of LOY in this cohort is low (5.1%). Considering that the average age of male population in SCOURGE cohort is 64.34, this rate is much lower than this in UK Biobank (Thompson et al., 2019, Nature), which also used a cutoff of 10% for clonal LOY. The authors should discuss this issue in discussion

6. Many previous studies have demonstrated that LOY is associated with mCAs in healthy individuals and cancer patients (summarized in Guo et al., 2021, Mutation Research Reviews). However, Perez-Jurado et al. found there is no significant correlation between the presence of LOY and CME (mCAs). Particularly, the authors found there is a high OR underlying the association of gonosomal aneuploidies with the presence of CME (mCAs). Given that LOY is also a condition of gonosomal aneuploidy, it is important to explain why LOY is not associated with the presence of CME (mCAs). The authors should discuss the reason for this inconsistent result.

7. A reference for applying immunosenescence to explain the pathogenetic mechanism of mCAs and LOY is needed. This theory has been summarized in a recent review (PMID: 33866012).

8. A suggestion: it will be interesting to determine whether CME is also associated with transcriptomic biomarkers of immune dysfunction and deficient response to SARS-COV2 infection? This analysis can be done if data is available.

9. A suggestion: it will be interesting to see whether the LOY+ COVID-19 patients that have recovered display a decreased LOY fraction, because it will give us some clues to understand the complexity in the cause-and-consequence relationship between LOY and COVID-19. This analysis can be done if data is available.

10. Indeed, clonal hematopoiesis (CH) is recently found to be associated with the risk for severe COVID-19. A recent publication was missed (Bolton et al., 2021, Nature Communications). Moreover, a recent study from Forsberg revealed that CH and LOY can co-exist in peripheral blood of the same healthy individuals (Ljungstrom et al., 2021 Leukemia). So, the authors should consider whether the increased mortality of COVID-19 in LOY+ cases was really explained by the presence of LOY or can also explained by the co-existence of CH.

We appreciate very much the constructive comments, criticisms, and support of the reviewers for our work, as well as their suggestions that have clearly contributed to improve the manuscript. Please find below the response to the reviewer's comments point by point.

Reviewer #1 (Remarks to the Author):

1. Fig 1: the authors may wish to separate the data of sex-biased hospitalization & mortality rate and estimated prevalence of CME and LOY in men into different figures as the latter is only from the male population. Or a gender-specific style of combined figure is recommended.

The objective with this figure was to show the sex-biased hospitalization and mortality by age, and how mCAs/CME (affecting both sexes) and LOY (affecting only males) somehow parallels and may contribute to those curves. We have done a gender-specific style of combined figure as recommended by the reviewer.

2. A table of demographic parameters comparing control and LOY patients is needed.

We have added the information about sex and average age by group in table 1. We are not aware of any additional (and available) demographic parameter that may add relevant information.

3. It is surprising that the authors analyzed cell-type composition and functionality by whole blood RNA transcriptomic analysis using, which is a low-resolution research method and might cause over-interpretation of the data, especially the sample size is relatively small (LOY n=11, Control n=32). i.e., it is hard to imagine that the increased counts of endothelial cells were suggested without any other direct quantitative method. Further, transcriptomic and functional analyses of sorted blood cells is recommended for demonstrating the mechanistic basis.

We agree that whole blood RNA transcriptomic analysis is a relatively low-resolution research method to infer cell-type proportions and might cause over-interpretation of the data, especially with small sample sizes. However, the interpretation and estimation of cell type content on the provided sample based on bulk transcriptome analysis with several methods has been proven to be quite accurate (PMID: 32332754, PMID: 30670690, PMID: 30670690, PMID: 31061481) and has been used thereafter in multiple studies. Of course, we also fully agree with the reviewer that transcriptomic and additional functional analyses of sorted cells in individuals with LOY might add quite relevant information on physiopathogenic mechanisms for disease susceptibility, but we consider that those studies are out of the scope of this manuscript. Our data are based on the analysis of samples previously collected and stored at EGCUT (retrospective), with no additional prospective sampling done. We do not have currently easy access to new samples from patients with LOY. In any case, transcriptomic analyses from sorted single cells of individuals with LOY have been reported elsewhere (Dumanski et al Cell Mol Life Sci. 2021 Apr;78(8):4019-4033).

4. Fig 3: Although the prevalence of LOY in males was considerably higher compared to the prevalence of detectable CME in the general population, the 90-day risk of COVID19 lethality was higher in CME than LOY (OR 1.78 vs. 1.40). Therefore, listing the contribution of gender-specific CMEs to COVID 19 lethality would provide a direct

comparison between the contributions of male CME and LOY to COVID19 lethality.

We have shown the requested data separated by sex in the new figure 3 as suggested, and further discussed in the manuscript. In fact, the contribution of CMEs/mCAs to COVID 19 lethality are stronger and more significant in males only (OR 2.16; 95%CI: 1.19-3.93). Although in the same direction, the split sample size was not enough to achieve statistical significance for an association in females (OR 1.32; 95%CI: 0.66-2.67). Please see the section “Association between mCA and COVID-19 severity”.

Reviewer #2 (Remarks to the Author):

The paper from Perez-Jurado et al. carried out a LOY analysis in a large cohort of 9578 patients with COVID from SCOURGE study. They revealed three important findings: 1) the presence of CME in both sexes was significantly associated with COVID-19 lethality; 2) the presence of LOY in male was significantly associated with COVID-19 lethality; 3) LOY was associated with dysregulated transcriptomic of immune function that involved in response SARS-CoV2 infection. Overall, this study is very interesting because it provides evidence that host genetics underlie at least part of the sex-biased severity and mortality of COVID-19. I am supportive of its publication and I have several comments to this paper as follows.

We appreciate the positive comments and support for our work.

1. I think the author should keep the use of terms and abbreviations along with the previous studies. For example, the authors use chromosomal mosaic events (CME) to refer autosomal and X alterations. However, the use of mosaic chromosomal alterations (mCAs) has spread in the field (e.g., PMID: 29995854; 32581363; 32581364; 34099924; 33866012). So, I suggest the authors to use mCAs and rephrase throughout.

We thank the reviewer for this comment and suggestion. We are fully aware that both terms and abbreviations, chromosomal mosaic events (CMEs) and mosaic chromosomal alterations (mCAs), have been used by us and others in the previous literature to refer to identical events. Although CME was the terminology and abbreviation used following the seminal manuscripts about the topic in 2010-2012, we agree that the use of mosaic chromosomal alterations (mCAs) has spread in the field. Therefore, we accept the editorial indication in this regard, and we have changed the abbreviation throughout the manuscript and figures.

2. In addition, the terms X chromosome monosomy (XCM) and LOY are overlapped. To avoid confusion, I suggest the authors to use LOY alone, because LOY can mean XCM in males.

We respectfully disagree with this additional comment. X chromosome monosomy (XCM) can be the consequence of somatic loss of chromosome Y (LOY) in XY males, but it can also result from meiotic gonosomal non-disjunction, from somatic loss of chromosome X (LOX) in XX females and from other events, so the terms XCM and LOY are not completely equivalent. In fact, in addition to the 226 males with mosaic LOY, we report in the current manuscript five female individuals with mosaic XCM with or without additional mCAs, two with mosaic X0/XX (not LOY) and three with mosaic X0/XY likely due to early somatic LOY during embryogenesis (table S3). The

use of LOY (or mLOY) to refer to females would not be appropriate. Therefore, we think that using both terms and abbreviations in the manuscript does not add confusion but the opposite.

3. Age is the biggest risk factor for COVID-19 in both males and females. As the authors have shown, individuals with CME have an average age (79.5 yrs) much higher than non-CME controls (70.6 yrs) and this is also the case for LOY male (83.5 yrs) and non-LOY males (69.1). So, I think the results that the presence of CME and LOY are significantly associated with COVID-19 lethality may be confounded by age and age should be adjusted.

It was already done in the initial version of the manuscript, and we have further clarified it in the text and figure. All values were adjusted by age and sex.

4. Since CME has a higher OR value than LOY for 90-day COVID-19 lethality, it is interesting to consider sex in the analysis of the association of CME with 90-day COVID-19 lethality.

As explained in the response to reviewer 1, we have shown the requested data separated by sex in the new figure 3 and Table 1 as suggested, and further discussed in the manuscript. The contribution of CMEs/mCAs to COVID 19 lethality are stronger and more significant in males only (OR 2.16; 95% CI: 1.19-3.93). Although in the same direction, the split sample size was not enough to achieve statistical significance for an association in females (OR 1.32; 95% CI: 0.66-2.67). Please see the section “Association between mCA and COVID-19 severity”.

5. I noted that the prevalence of LOY in this cohort is low (5.1%). Considering that the average age of male population in SCOURGE cohort is 64.34, this rate is much lower than this in UK Biobank (Thompson et al., 2019, Nature), which also used a cutoff of 10% for clonal LOY. The authors should discuss this issue in discussion

Considering the mean age of males in our study is 64 years, the prevalence of clonal LOY is within the reported range in the different publications with the cutoff of 10%: 5–10% of men around 60 (15–20% at 70 and 20–40% at 80, respectively). Our data are also consistent with our own previous results (Zhou et al Nat Genet 2016, González et al BMC Bioint 2020), also based on SNP array data.

The greatest discrepancy comes from data generated by another group, and it is especially remarkable in two manuscripts that refer to the same data from the UK biobank population analyzed by two different groups (Thompson et al., Nature 2019 vs Loftfield et al., Sci Rep 2018), as you can see in the figures below from these manuscripts (left and right, respectively).

Thompson et al discussed the issue in their manuscript, claiming their analytical method had much higher sensitivity to detect mLOY (numbers are larger than four-fold in very age interval in Thompson et al), and the main argument to validate their statement was the stronger and more significant associations with multiple presumed loci of susceptibility to LOY in their GWAS when using their method. However, when some of the same authors perform an association with cardiovascular mortality in a more recent manuscript (Sano et al, 2022), the study was done selecting only individuals with >40% of the cells with LOY, whose detection would be similar with any method.

We do not know which are the reasons for this significant discrepancy, but our data correspond to a different population obtained with similar SNP arrays. Anyway, we have previously shown that MADloy, used in this manuscript, is a robust calling method of LOY in comparison with the other methods previously used, using both simulations and real data (González et al BMC Bioinf 2020). We have briefly discussed this issue as suggested. Please see section “Association between LOY and COVID-19 severity”.

6. Many previous studies have demonstrated that LOY is associated with mCAs in healthy individuals and cancer patients (summarized in Guo et al., 2021, Mutation Research Reviews). However, Perez-Jurado et al. found there is no significant correlation between the presence of LOY and CME (mCAs). Particularly, the authors found there is a high OR underlying the association of gonosomal aneuploidies with the presence of CME (mCAs). Given that LOY is also a condition of gonosomal aneuploidy, it is important to explain why LOY is not associated with the presence of CME (mCAs). The authors should discuss the reason for this inconsistent result.

The association in other previous studies has never been strong. In fact, genetic factors predisposing to mLOY do not overlap much with those predisposing to mCAs. We also show here that there is not significant overlap among the deregulated genes in blood in both conditions (LOY and copy-neutral mCAs). However, the lack of significant association in our data might also be related to the relatively small sample size, compared with larger datasets.

7. A reference for applying immunosenescence to explain the pathogenetic mechanism of mCAs and LOY is needed. This theory has been summarized in a recent review (PMID: 33866012).

We have included the review article as a reference as suggested.

8. A suggestion: it will be interesting to determine whether CME is also associated with transcriptomic biomarkers of immune dysfunction and deficient response to SARS-COV2 infection? This analysis can be done if data is available.

We have taken extra time and effort to obtain these data. *A priori*, there is a big difference between individuals with LOY and those with mCA: the mosaic alteration in the clonal cell line is uniform in LOY but highly variable in mCAs (any autosomal fragment or chromosome). In fact, most genes deregulated in individuals with LOY are located on the Y chromosome. Therefore, we have selected individuals with copy neutral mCAs to minimize the effect of large mosaic copy number variants on transcriptome. The results of this transcriptomic analysis, as somehow expected, are more disperse and no significant correlation was seen between the top genes deregulated in patients with mCAs and those deregulated in response to SARS-COV2. These data have been included in the methods and results sections (Blood transcriptome in individuals with mCAs) with an additional supplementary table (Table S11).

9. A suggestion: it will be interesting to see whether the LOY+ COVID-19 patients that have recovered display a decreased LOY fraction, because it will give us some clues to understand the complexity in the cause-and-consequence relationship between LOY and COVID-19. This analysis can be done if data is available.

This information can be inferred as the association is stronger when we perform lineal analysis, as shown in the manuscript.

10. Indeed, clonal hematopoiesis (CH) is recently found to be associated with the risk for severe COVID-19. A recent publication was missed (Bolton et al., 2021, Nature Communications). Moreover, a recent study from Forsberg revealed that CH and LOY can co-exist in peripheral blood of the same healthy individuals (Ljungstrom et al., 2021 Leukemia). So, the authors should consider whether the increased mortality of COVID-19 in LOY+ cases was really explained by the presence of LOY or can also explained by the co-existence of CH.

We thank the reviewer for this comment and the recent reference associating clonal hematopoiesis with gene mutations with severe COVID19, which we have now included. We agree that the detection of CME/mCA and/or LOY in more than 10% of peripheral blood cells means that there is an underlying clonal hematopoiesis as already discussed, although it does not distinguish whether the chromosomal alteration is the cause (driver) or a consequence of the clonal expansion of the cells carrying it. We have included an additional paragraph further discussing these issues, as recommended.

“Clonal hematopoiesis of indeterminate potential due to expansion of peripheral blood cells with acquired point mutations in a specific set of genes, is a similar condition that also increases with age and associates increased risk for cancer, cardiovascular disease, and decreased overall survival.³⁵ A significant overlap after adjusting for age has been proven between detectable mCAs or LOY and mosaic gene mutations, suggesting a possible synergistic relationship between clonal hematopoiesis with gene mutations and acquired chromosomal rearrangements.³⁶ Interestingly, a relationship between clonal hematopoiesis with gene mutations and risk of severe infections, including severe COVID-19, has also been recently documented.³⁷ However, although the increased severity and mortality of COVID-19 could be explained in part by the co-existence of clonal hematopoiesis, the underlying mechanisms in

individuals with LOY are likely different from those in individuals with mCAs or mosaic gene mutations.”

REVIEWERS' COMMENTS:

Reviewer #1 (Remarks to the Author):

My concerns have been fully addressed.

Reviewer #2 (Remarks to the Author):

The authors have thoroughly revised the manuscript and satisfactorily addressed all concerns I raised. I enjoyed reading this paper and believe it is suitable for publication.